# ITGB3-mediated uptake of small extracellular vesicles facilitates intercellular communication in breast cancer cells

Pedro Fuentes [1,2], Marta Sesé [1,2], Pedro J. Guijarro[1,5], Marta Emperador [1,4,5], Sara Sánchez-Redondo[3], Héctor Peinado [3], Stefan Hümmer [1,2 ✉] & Santiago Ramón y. Cajal [1,2 ✉]

Metastasis, the spread of malignant cells from a primary tumour to distant sites, causes 90% of cancer-related deaths. The integrin ITGB3 has been previously described to play an essential role in breast cancer metastasis, but the precise mechanisms remain undefined. We have now uncovered essential and thus far unknown roles of ITGB3 in vesicle uptake. The functional requirement for ITGB3 derives from its interactions with heparan sulfate proteoglycans (HSPGs) and the process of integrin endocytosis, allowing the capture of extracellular vesicles and their endocytosis-mediated internalization. Key for the function of ITGB3 is the interaction and activation of focal adhesion kinase (FAK), which is required for endocytosis of these vesicles. Thus, ITGB3 has a central role in intracellular communication via extracellular vesicles, proposed to be critical for cancer metastasis.

[1] Translational Molecular Pathology, Vall d'Hebron Research Institute (VHIR), Universitat Autònoma de Barcelona (UAB), Barcelona, Spain. [2] Spanish Biomedical Research Network Centre in Oncology (CIBERONC), Madrid, Spain. [3] Microenvironment & Metastasis Group, Molecular Oncology Program, Spanish National Cancer Research Centre (CNIO), Madrid, Spain. [4] Present address: Tumor Biomarkers Group, Vall d'Hebron Institute of Oncology (VHIO), Barcelona, Spain. [5] These authors contributed equally: Pedro J. Guijarro, Marta Emperador. ✉email: stefan.hummer@vhir.org; sramon@vhebron.net

Metastasis accounts for 90% of cancer deaths. It is a multistage process culminating in colonisation of a new environment[1], and each step relies on the tumour cells' interactions with their microenvironment. Extracellular vesicles (EVs) secreted from primary tumours are key in this interaction[2,3]. EVs contain different molecular cargo (protein, RNA, or lipids) that can modify the local and distant environment, enabling primary tumours to evolve, establish pre-metastatic niches and metastasise[4,5].

EVs are a heterogeneous group of secreted membranous vesicles including microvesicles, ectosomes, and exosomes[6]. They have become valuable biomarkers in liquid biopsies, and existing research has focused on their characterization in different cancer types[7,8]. However, the mechanisms underlying their biosynthesis, release from donor cells, and uptake into target cells remain poorly understood. A major difference among vesicle subtypes is their origin: while microvesicles and ectosomes bud from the plasma membrane, exosome formation begins on early endosomes[9]. After maturation in multivesicular bodies through the invagination of the endosomal membrane and formation of intraluminal vesicles (ILVs), exosomes are released by fusion of ILVs with the plasma membrane[10]. The specific proteins incorporated into ILVs are regulated mainly by the ESCRT (endosomal-sorting complex required for transport) of which there are 4: ESCRT-0, I, II, and III. Interference with protein function within these complexes has been reported to block exosome biogenesis[11].

Once released from the donor cell, EVs induce cell signalling, either by interacting with target cell–surface proteins or being taken up into the receiving cell[6,9]. This uptake is currently the least understood step in vesicle-based intercellular communication, and proposed mechanisms range from passive membrane fusion to active uptake via macropinocytosis or endocytosis[12]. Interestingly, this bears similarity to the mechanisms described for virus endocytosis[13], which begins when viral membrane glycoproteins bind to glycoprotein attachment factors, such as heparan sulfate proteoglycans (HSPGs), on the target cell surface. HSPGs have also been demonstrated to be essential for uptake of EVs[14–16].

Integrins are multifunctional heterodimeric cell–surface receptor molecules that serve as entry receptors for a plethora of viruses. Uptake of herpes viruses has been reported to be dependent on αvβ3 integrin, and in Kaposi's sarcoma-associated herpesvirus (KSHV), binding of the virus to the cell surface via interactions with heparan sulfate results in temporary interaction between HSGPs and αvβ3[17]. This activates the integrin, enabling binding and activation of focal adhesion kinase (FAK) and subsequent assembly of endocytic machinery[17,18]. Interplay between HSGPs, integrins, FAK and endocytosis also regulates the turnover of focal adhesion sites[19–22]. This turnover allows the dynamic attachment of cells to surfaces and enables cell migration. Key for this process is integrin recycling, enabling local control of FAK activity. HSPGs of the syndecan family, particularly syndecan-4, play a crucial role in this process by regulating integrin recycling.

Early integrin studies in cancer research focused on their role as cell adhesion receptors, though new roles have since been uncovered, particularly in the regulation of metastatic processes by exosomes[23,24]. But while the role of integrins within exosomes has been studied, little is known about their role in acceptor cells. The fundamental role of integrins in tumour progression—particularly metastasis—has long been recognised, but their exact molecular functions remain incompletely understood[25]. We recently identified integrin beta 3 (ITGB3) as a factor involved in tumour stress resistance and demonstrated its essential role in metastatic progression of triple-negative breast cancer (TNBC)[26].

We—and others—have proposed that ITGB3 may be required for the formation of macro-metastatic foci[26,27]. Following on from this, we sought to investigate the mechanisms of cellular ITGB3 in exosome biogenesis and uptake in relation to tumour metastasis.

Our work now demonstrates that EV-stimulated clonal growth relies on ITGB3 in EV-receiving cells and that ITGB3 plays a central and thus far unknown role in EV uptake. The requirement for ITGB3 derives from the process of integrin endocytosis, allowing the internalisation of EVs captured by ITGB3-interacting HSPGs. This process is furthermore regulated by the ITGB3-interacting FAK, which is activated by EVs in an ITGB3-dependent manner and required for endocytosis of these vesicles. Thus, the transmembrane protein ITGB3 is required both for extracellular recognition of EVs by interacting with HSPGs and for intracellular FAK activation. The stimulated endocytosis of integrins, known from the process of focal adhesion disassembly, results in the coordinated cellular uptake of ITGB3- and HSPG-associated EVs. Thus, the central role of ITGB3 in intracellular communication via EVs and the proposed function of EVs in cancer metastasis might explain the requirement for ITGB3 in breast cancer metastasis.

## Results

**ITGB3 is required for EV-induced colony formation**. We previously demonstrated that ITGB3 is required for lung metastasis formation[26], but follow-up experiments revealed no defect in the initial homing of shITGB3 cells to the lung (Supplementary Fig. 1a–c). The next step in metastasis—formation of colonies within the distant organ—involves clonal growth from a single cell[28], which is dependent on factors secreted by other cells; in the laboratory, conditioned medium (CM) is generally used to generate clonal cell lines[29–32]. To assess the role of secreted factors on the colony-forming capacity of MDA.MB.231 cells, we measured the CM requirements for anchorage-dependent clonal growth (Fig. 1a, b). We seeded 500,000 cells, let them grow for 48 h, then collected the medium. As expected, in the presence of CM, the colony-forming capacity of MDA.MB.231 cells increased more than 2.5-fold, while shITGB3 cells had only a 1.3-fold increase (Fig. 1a, b). This difference can be attributed to the CM, since the colony-forming capacity of parental and shITGB3 MDA.MB.231 cells was not significantly altered after 7 days in normal growth medium (Fig. 1a, b). Similar results were obtained with a second shRNA construct, excluding a possible off-target effect (Supplementary Fig. 2). To test if this effect was due to soluble factors or EVs in the CM, we depleted the CM of EVs (100,000 $g$ spin)[33,34], and found that this, but not the removal of large EVs only (10,000 $g$ spin), completely negated the CM's increased colony-forming effect on the treated MDA.MB.231 cells (Fig. 1a, b; Supplementary Fig. 3a, b). Furthermore, the CM, but not the EV-depleted CM, derived from shITGB3 cells also increased the colony-forming capacity of MDA.MB231 cells. As in the case of MDA.MB.231-derived CM, this effect was not observed in shITGB3 cells. These results therefore indicate that ITGB3 is required for increased colony-forming capacity in vesicle-receiving cells only.

During metastatic dissemination and homing within a different organ, neoplastic cells are exposed to EVs derived from a variety of cell types. In the metastatic mouse model, we previously observed that the capacity of MDA.MB.231 cells to form metastatic colonies in the lung is reduced in shITGB3[26]. We therefore asked if, besides MDA.MB.231-derived EVs, EVs from lung tissue cells could also stimulate colony growth of MDA.MB.231 cells. We used lung-derived IMR90 fibroblasts: CM from

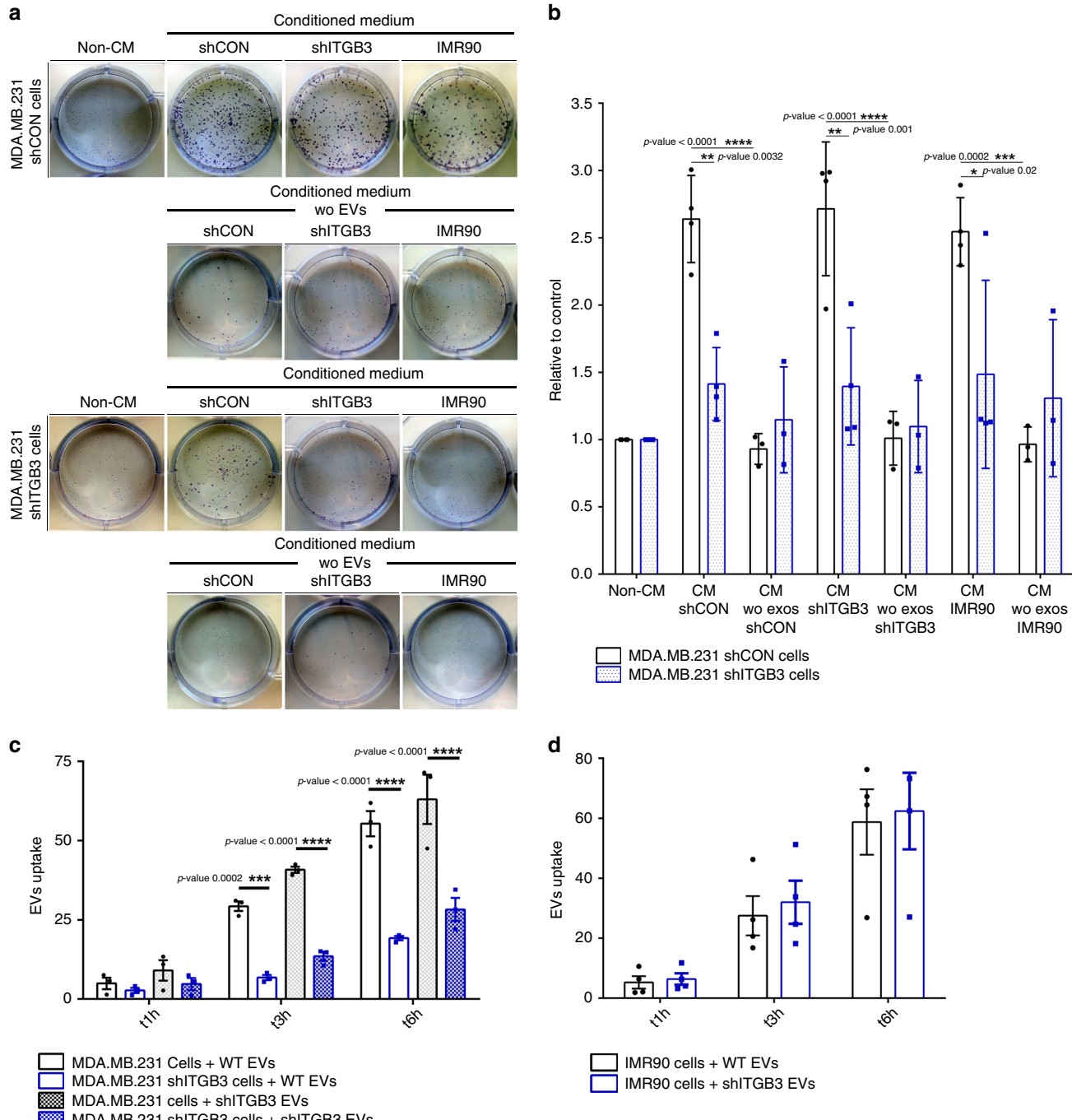

**Fig. 1 ITGB3 is required for EV-induced colony formation in MDA.MB.231 cells. a** Representative pictures are shown for each cell population and condition. **b** Conditioned medium (CM) was collected from exponentially growing MDA.MB.231 shCON, shITGB3, or IMR90 cells. In the case of vesicle-depleted CM (CM without exosomes), the same CM was split into two parts: one was used directly and the other was depleted of vesicles by ultracentrifugation before use, to establish the clonal cell growth response to the CM. Data are normalized to the non-conditioned control medium (DMEM). **c, d** Flow cytometry analysis for MDA.MB.231 shCON, MDA.MB.231 shITGB3 cells and IMR90 cells after incubating with 2–5 µg mL$^{-1}$ fluorescently labelled EVs derived from both MDA.MB.231 shCON and MDA.MB.231 shITGB3 cells at different time points. Source data are provided as a Source Data file.(*p value <0.05, **p value <0.01, ***p value <0.001, ****p value <0.0001; $n = 7$ for non-CM condition; $n = 4$ for CM shCON, CM shITGB3, and CM IMR90; $n = 3$ for CM wo EVs shCON and CM wo EVs IMR90. $n = 3$ for panel 1c, d). Data are represented as mean ± SD in (b–d). Statistical analysis including two-way ANOVA multiple comparisons was carried out using GraphPad Prism 6.01.

these cells, but not EV-depleted CM, stimulated colony growth in MDA.MB.231 cells in an ITGB3-dependent manner (Fig. 1a, b).

Taken together, these results show that the EV fraction of CM promotes clonal growth in MDA.MB.231 cells and indicates an essential role of ITGB3 in vesicle-receiving cells.

**ITGB3 is required for vesicle uptake**. An observed inability of shITGB3 cells to respond to the presence of EVs prompted us to speculate that these cells may have defective vesicle uptake and in turn defective delivery of the cargo molecules responsible for increased colony-forming capacity. To test this, we measured EV

uptake. EVs derived from shCON or shITGB3 cells were isolated by ultracentrifugation and fluorescently labelled with PKH26, and their uptake into both cell lines was measured by FACS at various time points after incubation. As shown in Fig. 1c, in the shCON cell line, EVs were taken up over time, and at 6 h up to 60% of cells were EV-positive. The portion of EV-positive cells (25% at 6 h) was lower in shITGB3 cells than in shCON cells. These results were confirmed by the use of an independent shRNA construct targeting ITGB3 (Supplementary Fig. 4). To exclude the possibility of a cell-line-specific function of ITGB3, we generated a stable shRNA-mediated knockdown of ITGB3 in MCF7 cells and demonstrated that EV uptake was also dependent on ITGB3 (Supplementary Fig. 5). Importantly, the origin of the vesicles—either from shCON or shITGB3 cells—did not affect their uptake into shCON and shITGB3 cells (Fig. 1c, d). These results were confirmed in IMR90 cells, which incorporated EVs from both cell lines with a similar efficiency (Fig. 1d). To further support the role of ITGB3 in EV uptake into receiving cells, we performed confocal microscopy of PKH26-labelled EVs on PFA-fixed and ITGB3 ($\alpha v\beta 3$)-stained cells. These experiments were carried out with shITGB3 cell-derived EVs to avoid staining of EV-localized ITGB3. Under these experimental conditions, we detected co-localisation between cellular ITGB3 and EVs at the surface of non-permeabilized cells (Supplementary Fig. 6).

These results demonstrate that the uptake of a significant portion of EVs relies on ITGB3 in the recipient cell. However, not all vesicles were dependent on ITGB3 (approximately 50%). This may have been due to incomplete knockdown of ITGB3, redundant pathways for vesicle uptake, or ITGB3 being required for the uptake of certain vesicle subtypes only. As we could not rigorously exclude any of these possibilities, we looked further at the mechanism of ITGB3-dependent EV uptake.

**ITGB3-dependent endocytosis of EVs requires HSPGs**. Having established that ITGB3 is fundamental for EV uptake in breast cancer cell lines, we explored the underlying mechanisms. First, we analysed how EVs might be captured at the cell surface. The described interactions of ITGB3 with other proteins include its selective binding to syndecans through the extracellular FERM domain[35–40]. Syndecans are a family of heparan sulfate-decorated cell–surface proteoglycans that recognise "heparin-binding" motifs. Treatment of different cancer cell lines with heparin, a highly sulfated glycosaminoglycan mimetic of heparan sulfate, has been shown to block EV uptake. To test if this occurred in MDA.MB.231 cells, we pretreated MDA.MB.231 shCON and MDA.MB.231 shITGB3 cells with fluorescently labelled vesicles derived from MDA.MB.231 cells. FACS analysis revealed that heparin blocked vesicle uptake to a similar degree as observed with shITGB3 knockdown. Furthermore, heparin treatment of shITGB3 cells had no additive effect on vesicle uptake, indicating that ITGB3 and HSPGs act on the same pathway (Fig. 2a). To strengthen our finding that HSPGs are required for EV uptake, cells were treated with heparinase prior to the addition of PKH26-labelled EVs. As occurred with heparin treatment, EV uptake was reduced by approximately 50% (Fig. 2b). As syndecans have been reported to directly interact with ITGB3, it is tempting to speculate that this interaction provides a physical link between exosomes and ITGB3.

Next, we asked how those captured vesicles are internalized. Previously proposed mechanisms for EV uptake and release of cargo into different target cell lines include fusion of vesicles with the plasma membrane and uptake of entire vesicles by endocytosis, phagocytosis or macropinocytosis[9,10,41]. We therefore began by exploring which mechanisms of vesicle uptake were used in MDA.MB.231 cells.

In line with previous reports[14,42,43], EV uptake at 4 °C was also severely blocked in MDA.MB.231 cells (Supplementary Fig. 7), suggesting that the underlying mechanism may be an energy-dependent process such as endocytosis. To test this, we blocked DYNAMIN activity using Dyngo-4a[44] and blocked clathrin activity using Pitstop-2[45]. As endocytosis is an essential cell process, inhibitor treatments were reduced to 30 min, followed by incubation with fluorescently labelled vesicles. Measurements of cell fluorescence intensity after 3 h revealed that vesicle uptake in MDA.MB.231 cells was significantly reduced in Dyngo-4a but not Pitstop-2 treated cells. Depletion of DYNAMIN 2 by shRNA (shDyn2) or the overexpression of a DYNAMIN 2 dominant negative mutant (Dyn2-44K) confirmed the results obtained for Dyngo-4a treatment (Fig. 2c–e).

Interestingly, the reduction of EVs in Dyngo-4a treated cells was accompanied by an increase in the surface abundance of ITGB3 ($\alpha v\beta 3$), as determined by FACS (Supplementary Fig. 8a). These results are in line with previous reports that examined recycling of integrins as a key mechanism to regulate their function at the cell surface[21]. In addition, we detected co-localization of ITGB3 ($\alpha v\beta 3$) with EEA1 on confocal microscopy, and ITGB3 was co-localized to internalized EVs derived from shITGB3 cells (Supplementary Fig. 9). Altogether, these results imply that not solely the presence of integrin beta 3 on the cell surface, but the active internalisation of the integrin in a DYNAMIN-dependent manner appears to be crucial for EV uptake.

As described above, the uptake of a significant fraction of EVs into MDA.MB.231 cells relies on ITGB3, HSPGs and DYNAMIN. To determine whether the remaining vesicles detected by FACS analysis were taken up into the recipient cells or attached to the cell surface, we included an additional wash step with citric acid-containing buffer (CAB wash) prior to FACS analysis[46]. In shCON and shITGB3 cells, CAB wash only slightly reduced the EV-derived fluorescence signal (Fig. 3a), indicating that vesicles do not accumulate at the cell surface before being taken up. Supporting this, heparin treatment, which already interferes with the capture of EVs at the cell surface, resulted in a similar reduction in EVs following the CAB wash (Fig. 3b). Furthermore, interference with the DYNAMIN-dependent internalisation of EVs through Dyngo-4a treatment did not result in accumulation of EVs at the cell surface (Fig. 3b). Thus, EV recognition and uptake appear to be intimately linked, preventing the accumulation of EVs at the cell surface when uptake is impaired.

To provide further evidence for these results independent of the FACS-based measurements, we assessed EV uptake into recipient cells on confocal microscopy. Cells treated with PKH26-labelled EVs were fixed, and alpha-tubulin staining of permeabilized cells was used for 3D reconstruction (Fig. 3c, d), to discriminate between surface-bound and internalized vesicles. Analysis of fluorescently labelled particles firstly showed a strong reduction in shITGB3 and shDyn2 cells (Fig. 3e). Furthermore, the ratio of surface-bound and internalized vesicles resembled our results obtained with the CAB wash in the FACS-based measurements of EV uptake (Fig. 2a). Our data therefore suggest that ITGB3 is essential for EV uptake in recipient cells. Furthermore, analysis of the PKH26-labelled EV uptake in shDyn2 cells revealed a strong reduction in the number of fluorescent particles (Fig. 3c–e); the proportion of surface-bound and internalized vesicles was similar to control and shITGB3 cells. Thus, FACS analysis combined with CAB-wash and confocal microscopy demonstrate that, when DYNAMIN-dependent EV uptake is impaired, EVs do not accumulate at the cell surface. Therefore, the interaction of EVs with the cell surface appears to be a highly dynamic process; if attachment of EVs is not intimately linked to endocytosis, EVs may detach again from the cell surface.

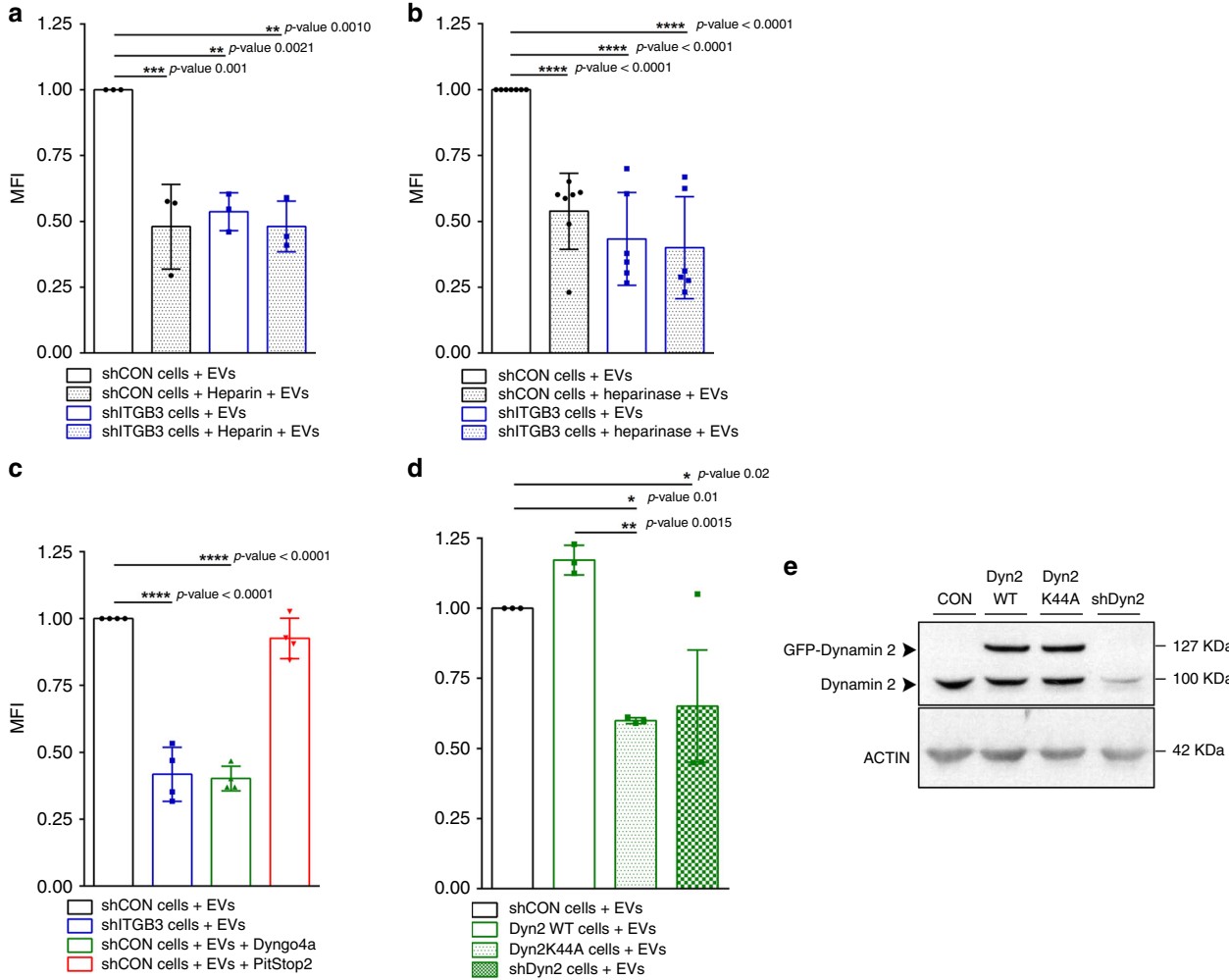

**Fig. 2 Endocytic uptake of EVs depends on ITGB3, HSPGs and DYNAMIN activity.** Internalization of 2–5 µg mL$^{-1}$ PKH26-labelled EVs derived from MDA.MB.231 cells by MDA.MB.231 shCON or MDA.MB.231 shITGB3 cells, measured by FACS. **a** Uptake of vesicles was measured after pretreatment with 10 µg mL$^{-1}$ heparin as a competitive inhibitor for cell–surface heparan-sulfated proteins for 30 min. **b** Uptake was measured in cells that were untreated or pretreated with 1.2 mIU mL$^{-1}$ heparinase I and 0.6 mIU mL$^{-1}$ heparinase III for 3 h before EV incubation. **c** Uptake of vesicles was measured after pretreatment with 20 µM Dyngo4a or 10 µM Pitstop 2 for 30 min in shCON cells. **d** Uptake of vesicles measured in MDA.MB.231 cells infected with Dyn2WT− Dyn2K44A− and shDyn2 constructs. **e** Representative immunoblot showing endogenous and exogenous levels of DYNAMIN 2 protein. Source data are provided as a Source Data file. (*$p$ value < 0.05, **$p$ value < 0.01, ***$p$ value < 0.001, ****$p$ value < 0.0001, $n = 3$ for panel 2a, d; $n = 4$ for panel 2c; $n = 7$ for panel 2b). Data are represented as mean ± SD in (**a**–**d**). Statistical analysis including two-tailed unpaired Student's $t$ test data was carried out using GraphPad Prism 6.01.

**ITGB3-dependent alterations in the EV secretome.** Having established the involvement of ITGB3 in the endocytosis-driven uptake of EVs, we wondered how this might be reflected in the secretome of shITGB3 cells. To test this, shCON and shITGB3 MDA.MB.231 cells were maintained in low-serum-containing medium (exosome-depleted 0.5% fetal bovine serum (FBS)) for 48 h and vesicles in the cell culture supernatant were isolated by differential ultracentrifugation (Fig. 4a). The vesicles were characterized by Nanosight and cryo-electron microscopy (Fig. 4) and their proteome was determined by liquid chromatography–mass spectrometry (LC–MS)/MS and Western blot (Fig. 5). The overall abundance of EVs, measured by Nanosight and normalized to the number of EV-producing cells, revealed no significant difference between shCON and shITGB3 cells (Fig. 4b–d). Similar results were obtained when the protein quantity in the EV fraction after ultracentrifugation was measured by bicinchoninic acid (BCA) protein assay (Supplementary Fig. 10c). However, we did detect clear differences in EV quantity when we analysed different

vesicle populations by size. In line with previous reports, vesicles between 150 and 200 nm comprised the peak fraction of MDA.MB.231 cells[47]. This vesicle population was reduced in shITGB3 cells, while those measuring 50–125 nm were significantly increased (Fig. 4b, c) (Supplementary Fig. 10a–c). We confirmed these results on cryo-electron microscopy (Fig. 4d). In summary, vesicles in the supernatant of shITGB3 displayed an altered size distribution.

To test if these changes were accompanied by alterations in the proteome of the respective EVs, we performed LC–MS/MS. We identified 3204 proteins(Supplementary Data 1), 1195 of which exhibited a statistically significant difference in abundance between EVs isolated from shCON and shITGB3 cells ($q$.value < 0.05) (Fig. 5a, b). The fold change in protein abundance was calculated using the shCON condition as a reference, i.e., log2(shITGB3/shCON). The heat map in Fig. 5a summarizes the results from the independent biological replicates and highlights the striking differences in the

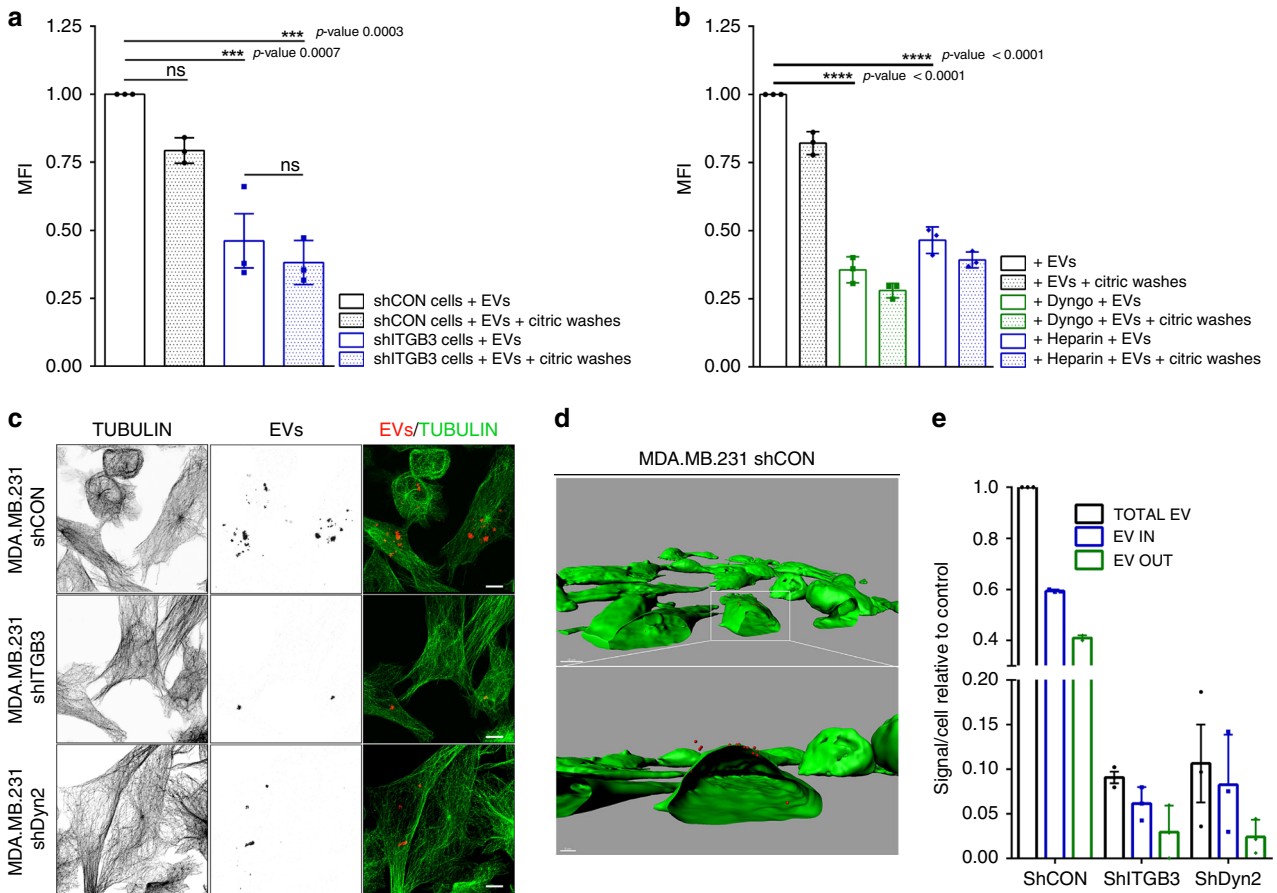

**Fig. 3 Interaction of EVs with the cell surface is a highly dynamic process.** Internalization of 2–5 μg mL$^{-1}$ PKH26-labelled EVs derived from MDA. MB.231 cells by MDA.MB.231 shCON or MDA.MB.231 shITGB3 cells, measured by FACS. **a** Uptake of vesicles was measured by FACS after subjecting the cells to acid washing. **b** Internalization of 2–5 μg mL$^{-1}$ PKH26-labelled EVs derived from MDA.MB.231 cells by MDA.MB.231 WT cells, pretreated with 20 μM Dyngo4a and pre-treated with 10 μg mL$^{-1}$ heparin, with or without subjecting the cells to acid washing. **c** Representative confocal pictures used for 3D reconstruction. **d** Representative 3D reconstruction picture of MDA.MB.231 shCON cells. Bar represents 5 μm. **e** Analysis of EV internalization in MDA. MB.231 shCON, shITGB3 and shDyn2 cells after 3D reconstruction. Source data are provided as a Source Data file. (***$p$ value < 0.001, ****$p$ value < 0.0001, $n = 3$). Data are represented as mean ± SD in (**a**, **b**, **e**). Statistical analysis including two-way ANOVA multiple comparisons was carried out using GraphPad Prism 6.01.

proteome of vesicles derived from shCON and shITGB3 cells. These differences were furthermore supported by GO-term and KEGG pathway analysis (Supplementary Fig. 11).

To validate our findings, we performed Western blot analysis on proteins from the EV fractions derived from shCON and shITGB3 cells (Fig. 5c). This confirmed the knockdown of ITGB3 in EVs from the shITGB3 cell line, accompanied by a reduction in ITGAV, the alpha subunit of the αvβ3 heterodimer of ITGB3. However, the levels of ITGB1, also described to be present in EVs, were almost indistinguishable between vesicle fractions derived from shCON and shITGB3 cells. These results not only confirm the specificity of our knockdown, but also indicate that different integrins might be associated with different types of EV. In line with our proteomic analysis, we also confirmed a strong reduction in TSG101, a member of the ESCRT family, and the tetraspanin CD81, a bona fide exosome marker. Importantly, this defect appears to be specific to exosomes, as general EV marker proteins such as flotillin-1 or actin were not affected by the knockdown of ITGB3. In contrast, certain proteins, such as EIF4E, were exclusively associated with vesicles derived from shITGB3 cells. These results were confirmed with another shRNA construct, excluding the possibility of an off-target effect (Supplementary Fig. 4). Furthermore, we obtained the same results in MCF7 cells (Supplementary Fig. 12), excluding a cell-

line-specific phenomenon. Finally, we performed iodixanol gradient purification of EVs isolated by ultracentrifugation from shCON and shITGB3 cells. Western blot analysis confirmed that while flotillin and actin were equally present in the EV fractions (fraction 6/7) from shCON and shITGB3 cells, CD81 and TSG101 were strongly reduced in those fractions from shITGB3 cells (Fig. 5d, e).

As described above, EVs from both MDA.MB231 shCON and shITGB3 cells were equally capable of promoting colony growth of recipient MDA.MB231 cells in an ITGB3-dependent manner. However, the striking differences in the EV secretome of shCON and shITGB3 cells prompted us to speculate that other cellular processes might be altered. To test this, we used an indirect coculture system[48] to measure the migratory capacity of IMR90 cells in the presence of MDA.MB.231 shCON or shITGB3 cells. In line with the reported pro-migratory effect of MDA.MB231-derived exosomes[49], IMR90 cells cocultured with MDA. MB.231 cells displayed an increased migratory capacity. Such migration was not induced in IMR90 cells cocultured with shITGB3 cells (Supplementary Fig. 13a, b). These results were not due to impaired uptake of vesicles derived from shITGB3 cells, since isolated vesicles from both shCON and shITGB3 cells were incorporated into IMR90 cells with similar kinetics (Fig. 1d). Thus, while secreted factors from shITGB3 cells did not promote

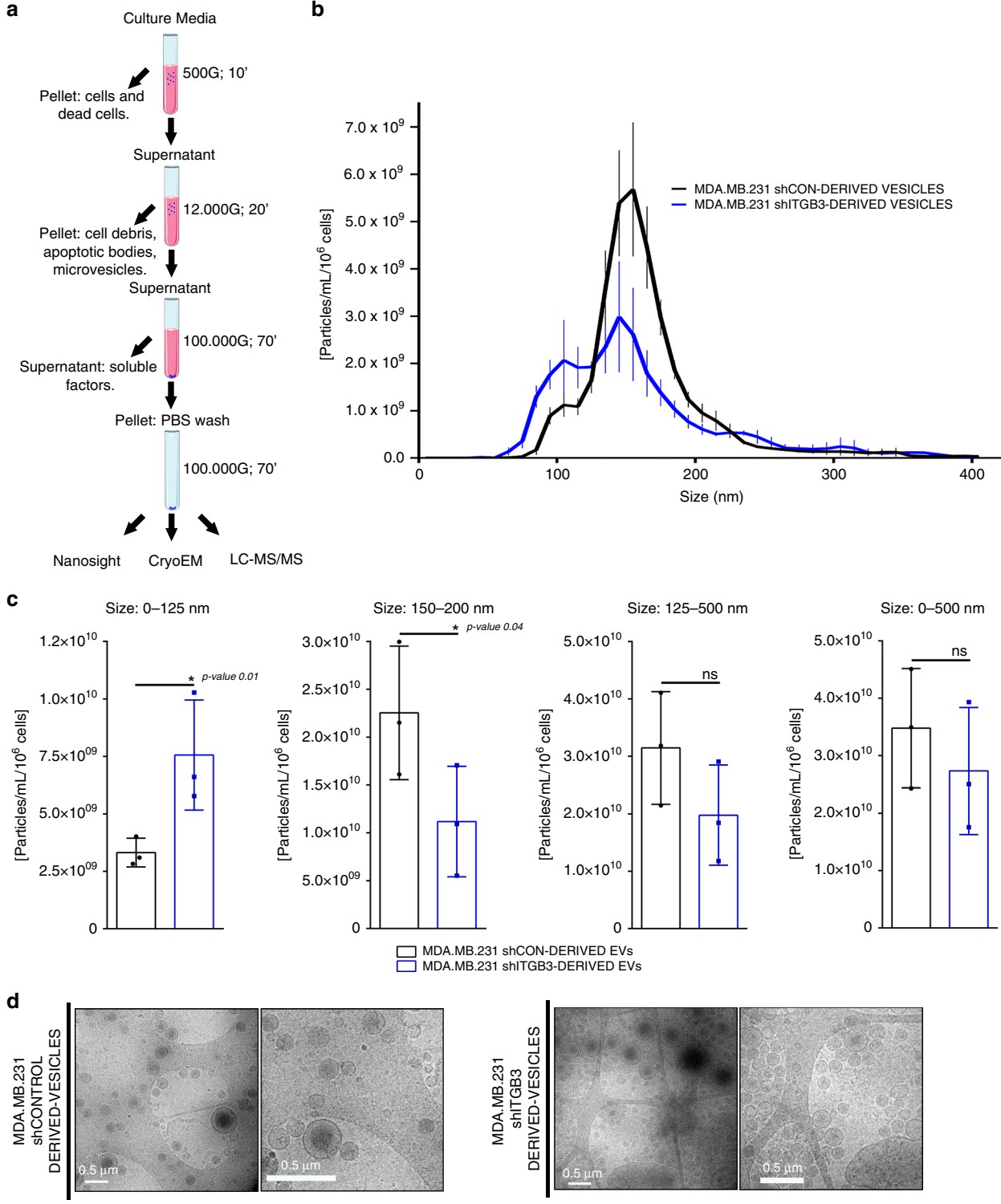

**Fig. 4 Characterization of EVs isolated from shCON and shITGB3 cells. a** Schematic workflow for the preparation of EV samples for LC–MS/MS analysis. **b** Vesicle size distribution based on NTA data. **c** Particle concentration of isolated vesicle fractions normalized per million cells. **d** Representative CryoEM images of EV isolated from cultured MDA.MB.231 shCON and MDA.MB.231 shITGB3 cells. Source data are provided as a Source Data file. Scheme created with BioRender (https://biorender.com/). (*$p$ value < 0.05, $n = 3$). Data are represented as mean ± SD in (**b**, **c**). Statistical analysis including two-tailed unpaired Student's $t$ test data was carried out using GraphPad Prism 6.01.

cell migration in IMR90 cells, MDA.MB231 colony growth was stimulated similarly by EVs from shCON and shITGB3 cells, indicating functional diversity within the heterogeneous population of EVs.

In summary, the overall number of vesicles was not significantly altered between shCON and shITGB3 cells but the EVs isolated from the cell culture supernatant differed in size and protein composition. Particularly exosomes, defined as

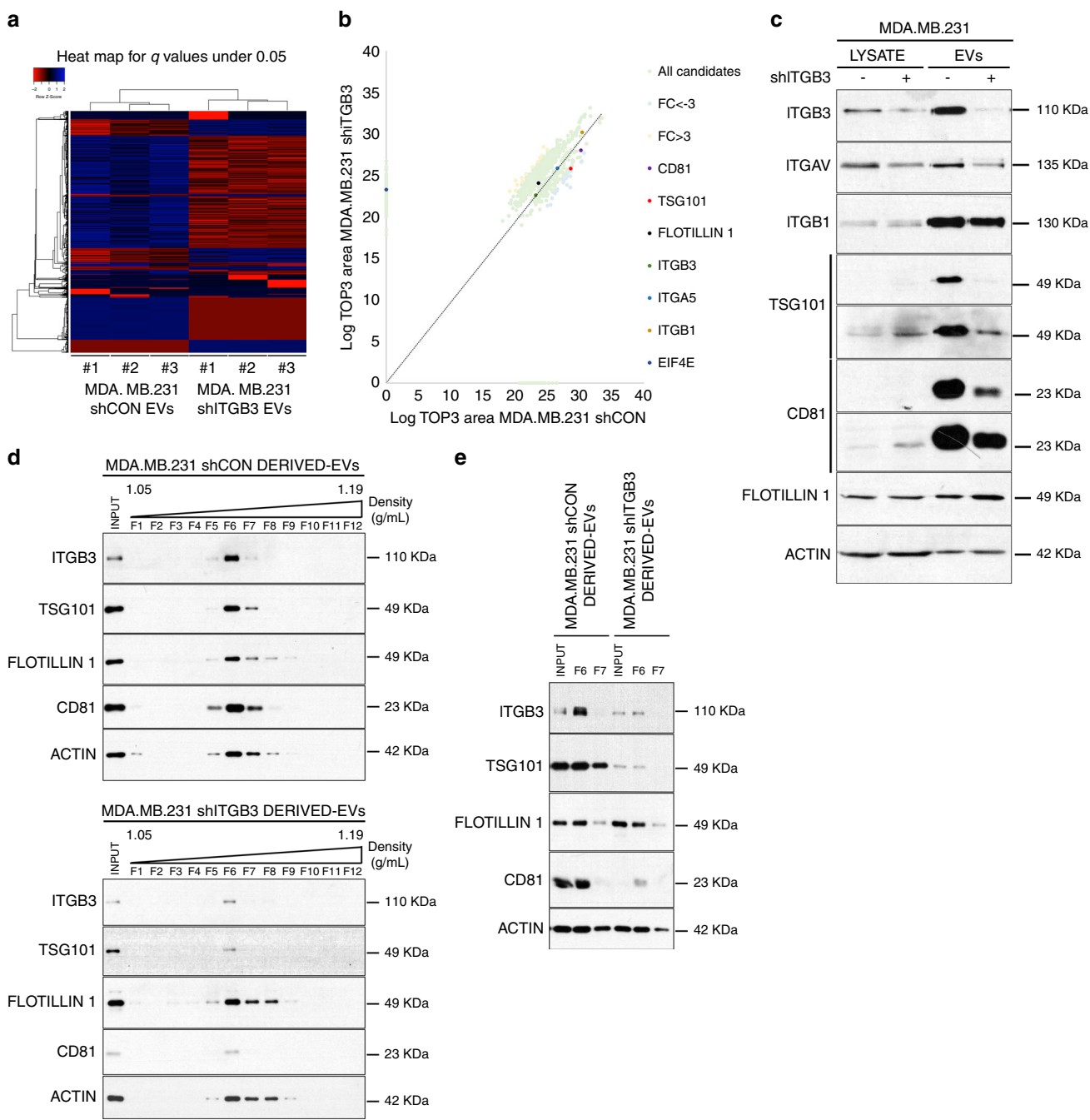

**Fig. 5 Alterations in the proteome of shITGB3-derived EVs. a** Heat map representation of the normalized protein abundance (Log TOP 3 area) (*q*.value < 0.05) for the comparison of experimental replicates. **b** Scatter plot comparison of the abundance (Log TOP3 area) of proteins in the LC–MS/MS analysis of MDA.MB.231 shCON− and MDA.MB.231 shITGB3-derived EVs. The proteins validated by Western blot are highlighted. **c** Representative immunoblot analysis of candidate proteins identified in the proteomic survey. **d** Representative immunoblots after separation of vesicles in an iodixanol gradient for both MDA.MB.231 shCON (upper panel) and MDA.MB.231 shITGB3 (lower panel). **e** Representative immunoblot analysis of input and fractions 6 and 7. Source data are provided as a Source Data file.

CD81/TSG101-positive vesicles, appear strongly diminished in the case of shITGB3. The cause of this phenotype might be due to defective exosome biogenesis in shITGB3 cells or linked to the role of ITGB3 in vesicle uptake. Regarding a putative role in exosome biogenesis, it is important to mention that the ITGB3-interacting protein SRC has previously been shown to be required for exosome biogenesis[34]. Linking the described alterations in the EV secretome of shITGB3 cells to the role of ITGB3 in EV uptake, the abundance of small vesicles (50–125 nm) indicates

that their uptake is particularly dependent on ITGB3. As we could not rigorously rule out if defects in exosome biogenesis or EV uptake were responsible for the alterations in the EV secretome of shCON and shITGB3 cells, we looked further at the mechanism of ITGB3-dependent EV uptake.

**Uptake of EVs by MDA.MB.231 cells is dependent on FAK.** As described above, DYNAMIN-dependent endocytic recycling is integral to the regulation of integrin function[50]. In the case of

ITGB3, this recycling enables the assembly and disassembly of focal adhesions to allow cell migration. A key regulator of this process is FAK, which binds to and regulates the function of both proteins: DYNAMIN and ITGB3[51–53].

To elaborate a putative link between FAK and ITGB3 in EV-related functions, we first monitored the activity of FAK in cells in different growth conditions. We prepared protein extracts from cells grown as colonies in the absence or presence of CM and normally growing cells, and monitored FAK activity by Western blot using the Y397 autophosphorylation as readout. In comparison to normal growth conditions, FAK activity was strongly increased when cells were grown as individual clones, particularly under CM-stimulated conditions (Fig. 6a). This FAK activity correlates with its described role in clonogenic growth[54,55]. However, FAK as a multifunctional protein kinase is required for clonogenic growth under basal conditions, complicating further analysis to establish a link to the ITGB3-dependent function in CM-stimulated colony formation.

Nevertheless, the described interaction between FAK and ITGB3 prompted us to examine in more detail if FAK activity is also required for EV uptake. To test this, cells were pre-treated with the small-molecule inhibitor FAK14 for 16 h to inactivate the kinase, then incubated with fluorescently labelled EVs. After 3 h, FACS analysis of internalized vesicles revealed a reduction in fluorescent intensity of approximately 60% compared to cells treated with DMSO as solvent control (Fig. 6b). Knockdown of FAK by siRNA confirmed these results (Fig. 6b, c). These data suggest that FAK activity is required for EV uptake into recipient cells. To test if FAK activity alone was sufficient for EV uptake in the absence of ITGB3, we transiently expressed FAK-GFP or the pCDNA3-GFP vector as a negative control in shCON and shITGB3 cells. Cells were treated with PKH26-labelled EVs for 3 h and the uptake into GFP-positive cells was determined by FACS. While we did observe a slight increase in EV uptake in shCON cells expressing FAK-GFP, the defect in vesicle uptake in shITGB3 cells was similar to the vector control (Fig. 6d). The activity of the overexpressed FAK-GFP was confirmed by Western blot, using the autophosphorylation of FAK on Y397 as readout (Fig. 6e). These results therefore suggest that, while FAK is required for EV uptake, ITGB3 interaction with HSPGs is required to link vesicle recognition at the cell surface to FAK activation in the cytoplasm.

As described above for the knockdown of ITGB3, a significant portion of vesicles were still taken up following FAK14 treatment or knockdown of FAK by siRNA. We therefore wondered if FAK inhibition would result in the same alterations in EVs in the cell culture supernatant as we observed in shITGB3 cells. To test this, control and FAK-14-treated MDA.MB.231 cells were maintained in low-serum-containing medium (exosome-depleted 0.5% FBS) for 48 h, and vesicles in the cell culture supernatant were isolated by differential ultracentrifugation (Fig. 4a). Western blot analysis of the obtained EVs revealed that CD81 and TSG101 were significantly reduced, while flotillin-1 and actin remained unaffected (Fig. 6f).

Considering these parallelisms in the EV-related functions of ITGB3 and FAK, we next wondered if ITGB3-mediated recognition of EVs was directly linked to the activation of FAK. To demonstrate this, we first serum-starved cells for 24 h to eliminate residual FAK activity[19,52], then added purified EVs. As shown in Fig. 6g, the purified vesicles resulted in FAK activation, as demonstrated by autophosphorylation at Y397. The specificity of this phosphorylation event was confirmed by FAK-14 treatment. Importantly, this activation was strongly reduced in shITGB3 cells, pointing towards an ITGB3-dependent activation of FAK. Moreover, FAK activation was

blocked when DYNAMIN-dependent endocytosis was inhibited (Dyngo), but not when clathrin-dependent endocytosis was inhibited (Pitstop) (Fig. 6g). Notably, among all tested signalling pathways reported to be downstream of ITGB3, only FAK activity was stimulated by EVs (Supplementary Fig. 14). Together with our previous observation that vesicle uptake relies on DYNAMIN-dependent endocytosis (Fig. 2b, c), these results suggest that DYNAMIN and FAK are mutually dependent and together regulate EV uptake in an ITGB3-dependent manner. Notably, DYNAMIN and FAK have been shown to be interdependent for regulating focal adhesion turnover[51,52], a scenario reminiscent of our observations.

In summary, we postulate that the transmembrane cell-surface protein ITGB3 plays a central role in the uptake of a subset of EVs. In this function, ITGB3 does not act as an EV receptor but is critical for mediating the link between EV capture at the cell surface and their uptake by endocytosis. Through the critical position of ITGB3 at the interface of extracellular and cellular interactions, the recognition of EVs at the cell surface by ITGB3-interacting HSPGs can be transmitted to the cytoplasm to trigger the interplay between DYNAMIN and FAK which ultimately allows the endocytic uptake of EVs (Fig. 6h).

## Discussion

Building on the findings[26,27,56] that ITGB3 is required for lung metastasis in MDA.MB.231 cells, we have demonstrated that the underlying mechanism may relate to the role ITGB3 plays in EV uptake and exosome biogenesis. In light of the preserved lung-homing capacity of shITGB3 cells in animal models, we focused on the next step in metastasis—clonal growth—and found that it was largely dependent on secreted factors in the cellular environment; shITGB3 cells had reduced EV uptake, rendering them insensitive to these factors. Our data indicate that this need for ITGB3 may be due to its dual role in connecting the capture of vesicles at the cell surface by HSPG-modified proteins like syndecans to the local activation of FAK-induced DYNAMIN-driven endocytosis (Fig. 6h).

As depicted in our model, endocytosis of EVs is linked to integrin internalisation. Integrins are established key players in the interaction between cells and their environment. This interaction must be tightly regulated for normal cell development; the deregulated expression of various integrins correlates with disease progression in several cancers. As transmembrane cell–surface receptors, integrins are critically positioned for cellular–extracellular interactions. Together with other scaffold proteins and signalling molecules they form focal adhesion complexes that link the actin cytoskeleton to the extracellular environment. These complexes form following integrin clustering induced by interactions with the extracellular matrix. The attachment of EVs to integrin-linked cell-surface receptors may induce a similar effect. The existence of such "endocytic hotspots[57]" for EV uptake on filopodia has been described previously. Cell spreading and motility requires active turnover of cell adhesion complexes, achieved mainly through endocytic recycling of integrins, a process that has been studied in detail in recent years[21,58]. Many common molecular players are involved, and certain specific integrin variations have been described, mainly relating to the highly variable C-terminal tail of beta integrins[59,60]. These sequence motifs define specific interactions with proteins, determining whether the uptake will be dependent upon caveolae, clathrin, neither, of both. In the case of ITGB3, endocytosis has been reported to rely solely on DYNAMIN[49]—our findings support this. Finally it is important to mention that the accurate localisation of individual EVs and the discrimination between different vesicle types among the heterogeneous population

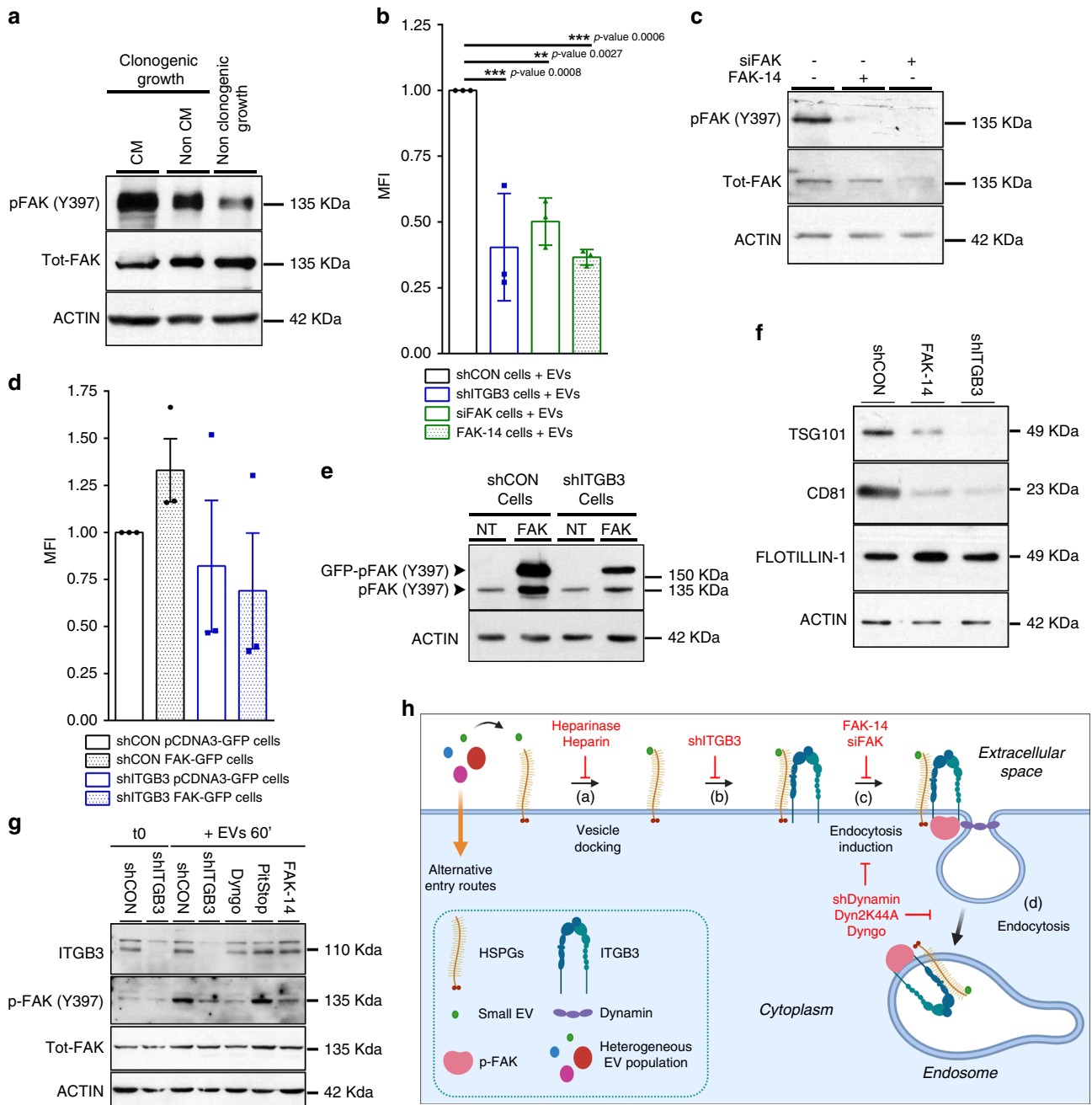

**Fig. 6 EV-induced activation of FAK is required for ITGB3-dependent EV uptake. a** Representative immunoblot analysis of pFAK activity either in normal conditions or clonogenic growth conditions with and without conditional medium (CM). **b** Internalization of 2–5 µg mL⁻¹ PKH26-labelled exosomes derived from MDA.MB.231 cells by MDA.MB.231 shCON, shITGB3, siFAK or shCON pretreated with 15 µM FAK 14 for 14–16 h, measured by FACS. **c** Representative immunoblot analysis confirming FAK knockdown and pFAK activity reduction after FAK-14 treatment. **d** Internalization of 2–5 µg mL⁻¹ PKH26-labelled exosomes derived from MDA.MB.231 cells by MDA.MB.231 shCON pCDNA3-GFP, shCON FAK-GFP, shITGB3 pCDNA3-GFP, and shITGB3 FAK-GFP, in GFP-positive cells measured by FACS. **e** Representative immunoblot analysis of pFAK activity in shCON pCDNA3-GFP, shCON FAK-GFP, shITGB3 pCDNA3-GFP, and shITGB3 FAK-GFP. **f** Representative immunoblot analysis of TSG101, CD81 and flotilin-1 on EVs derived from untreated MDA.MB.231 shCON or MDA.MB.231 shITGB3 cells or from control or FAK-14 inhibitor (15 µM) treated MDA.MB.231 shCON cells. **g** Representative immunoblot analysis of the autophosphorylation of FAK at Y397 as readout for its kinase activity. MDA.MB.231 shCON or shITGB3 serum-starved cells were pretreated with 20 µM Dyngo4a, 10 µM Pitstop 2 or 10 µg mL⁻¹ heparin for 30 min or with 15 µM FAK-14 for 14–16 h followed by addition of EVs for 60 min. **h** Model for the proposed role of ITGB3 in vesicle uptake and exosome biogenesis. (a) EV-HSPG interaction; (b) αvβ3 recruitment; (c) pFAK-DYNAMIN recruitment to endocytosis complex; (d) Dynamin-mediated internalization of EVs and EE formation. Source data are provided as a Source Data file. Model created with BioRender (https://biorender.com/). (**p value < 0.01, ***p value < 0.001 n = 3). Data are represented as mean ± SD in (**b**) and (**d**). Statistical analysis including two-way ANOVA multiple comparisons was carried out using GraphPad Prism 6.01.

of EVs is still hindered by the current limitations of the applied microscopy techniques[10]. This is particularly important, when considering the role ITGB3 in the selective uptake of small EVs. While NTA can readily discriminate those different sized EV populations, the resolution limit of conventional confocal microscopy does not permit visualisation of those vesicles and to discriminate among EV subpopulations at reliable resolution in vivo. Current advancements of FACS (asymmetric-flow field-flow fractionation technology) and microscope-based techniques will complement the NTA based technologies in the future, which will allow the precise determination of vesicle size and localisation in vivo[10,61].

As described above, FAK and DYNAMIN are key players in the ITGB3-dependent endocytosis of EVs. Supporting this, the interaction between FAK and the C-terminal tail of ITGB3 is well established[60,62]. FAK-dependent phosphorylation and regulation of the effector protein DYNAMIN is required for the turnover of focal adhesions[19,51,52]. Interestingly, our studies indicate that FAK and DYNAMIN are mutually dependent on each other for the ITGB3-dependent uptake of EVs. This interdependency might be due to the reported direct interaction between the two proteins for the endocytic uptake of ITGB1. However, given the marked differences in the cytoplasmic tail of ITGB3 and ITGB1, detailed molecular analysis exploring the involvement of other adapter proteins or the reported interaction with signalling molecules like SRC will be required to delineate the exact molecular mechanisms. As for ITGB3, we cannot rigorously exclude the possibility that FAK might also be required for exosome biogenesis. While our data strongly support a role of FAK in the ITGB3-dependent uptake of EVs, a closer examination of the potential role of FAK in exosome biogenesis might present an interesting focus for future investigation.

Interestingly, our proposed model for ITGB3-dependent EV uptake closely resembles the uptake of herpes viruses. Viral endocytosis begins with the viral capsid binding to integrin-associated HSPGs at the cell surface. Following attachment and receptor clustering, activation of receptor tyrosine kinases (FAK, SRC) drive specific signalling pathways that activate membrane fission and fusion factors. To date, a variety of mechanisms have been described for EV uptake. A similar variety has been described for the uptake of different viruses (20–300 nm) into specific cell types. Drawing a further comparison between the cell-specificity of viruses and that of EVs, we demonstrated that a specific fraction of EVs (approximately 50% of the total vesicles) were taken up in an ITGB3-dependent manner. Since this effect was not increased by combining treatments (shITGB3 + heparin), we conclude that a certain type of EVs are taken up via this route. Analysis of the EVs in cell culture supernatant revealed a significant increase in vesicles with a diameter between 50 and 100 nm in shITGB3 cells, which might indicate that the uptake of those vesicles is particularly dependent on ITGB3. Interestingly, vesicles of this size (termed exomeres or P200 exosomes) are described to contain less CD81 than other EVs[63,64], which might explain why the relative increase in these vesicles in the total isolated EVs results in a reduction of CD81 in the cell culture supernatant of shITGB3 cells. However, detailed molecular analysis using recently developed techniques like asymmetric flow field-flow fractionation of EVs are required to determine the exact molecular identity of those vesicles[61]. Although we cannot rigorously exclude that the differences in the EV secretome of shITGB3 cells might be caused by alterations in exosome biogenesis, the data presented here on the role of ITGB3 in vesicle uptake and the parallelism in the phenotype with the ITGB3-interacting protein FAK rather suggest a selective defect in the uptake of small vesicles devoid of CD81. The recently discovered variety in EV size and molecular composition is reminiscent of the variety seen in viruses, for which different uptake mechanisms

have been described. Finally, certain receptor cell-specific adaptations to this mechanism may also occur, such as the use of different integrin receptors. The most likely candidates would be integrins that can interact with heparinised cell-surface receptors and contain a β-subunit capable of binding and activating FAK.

We found that ITGB3 knockdown rendered breast cancer cells insensitive to secreted factors in their environment, reducing their clonal growth capacity. Intercellular communication represents an important target for future cancer treatment strategies. Current approaches target cancer cells, yet the past decade of research has demonstrated the importance of the cell environment and intercellular communication in disease progression and treatment resistance: blocking these interactions seems a sensible strategy to improve current treatments.

## Methods

**Cell culture and reagents.** Breast cancer cell lines were purchased from the American Type Culture Collection (ATCC) and authenticated by DNA profiling using short tandem repeat (GenePrint® 10 System, Promega) at Genomics Core Facility, Instituto de Investigaciones Biomédicas Alberto Sols CSIC-UAM. MDA.MB.231 and MCF7 cells were maintained in Dulbecco's modified Eagle's medium (DMEM) (Invitrogen) supplemented with 10% heat-inactivated FBS (Life Technologies) and antibiotics (100 U mL$^{-1}$ penicillin, 100 µg mL$^{-1}$ streptomycin) (Life Technologies). Cells were maintained at 37 °C in a 5% $CO_2$ humidified incubator. The FBS used for exosome purification experiments was depleted of EVs by centrifugation at 100,000$g$ for 1 h 10 min at 4 °C. For the preparation of medium with EV-depleted 0.5% FBS, EV-depleted FBS was added to DMEM to a final concentration of 0.5% v/v.

**Cell culture migration assays in co-culture.** Cells were cultured in a double chamber coculture system allowing cell types to grow in close proximity to one another without mixing (TC insert for 24-well plate, PET membrane bottom, translucent, pore size 0.4 µm, sterile, non-pyrogenic/endotoxin-free, non-cytotoxic, 1 pc./blister (Sarstedt). The upper part is a transwell comprised of 24 cell culture inserts with a 0.4-µm pore size to avoid mixing the two cell types. The lower part is a well of the 24-well plate. IMR90 cells were plated in the lower culture 24-well plate area in triplicate. Either MDA.MB.231 shCONTROL or MDA.MB.231 shITGB3 cells were plated in triplicate in the upper inserts. After 48 h growing in shared medium, IMR90 reached a monolayer and were treated overnight with mitomycin C (5 µg mL$^{-1}$, Santa Cruz Biotechnology). Then, a wound was made in the monolayer with a pipette tip, the medium was replaced, and the cells were incubated in normal conditions. Pictures of the wounds were taken after 24 h, and wound closure was measured using ImageJ software.

**Cell-surface flow cytometry staining protocol.** For cell-surface levels, after 1 h treatment with heparin (Sigma Aldrich), Dyngo-4a (Abcam) or Pitstop-2 (Abcam) or after an overnight treatment in the case of FAK-14 inhibitor, cells were detached using enzyme-free phosphate-buffered saline (PBS)-based Cell Dissociation Buffer (Gibco). Cells were washed in PBS solution and incubated with the FITC-conjugated ITGB3 antibody at room temperature for 1 h in non-permeabilized conditions. Cells were washed in PBS solution, suspended in FACS buffer (EDTA 2.5 mM, 1% bovine serum albumin (BSA) in PBS), and analysed on a FACSCalibur instrument with integrated FACSDiva (BD Biosciences) v8.0.1 Software.

**Clonal cell growth assay.** Cells were seeded in six-well plates (500 cells/well) and cultured for 7–9 days at 37 °C and 5% $CO_2$. The supernatant was discarded and the cells were gently washed with 1 mL of PBS. Clonal cell growth was assessed following staining with 0.5% crystal violet in dd$H_2O$. The crystal violet staining of cells from each well was solubilized using 15% acetic acid and the optical density of the solution was measured.

**Cryogenic electron microscopy (CryoEM).** EVs collected by ultracentrifugation were analysed at similar dilutions on electron cryomicroscopy (CryoEM) in the microscopy facility at the Universitat Autònoma de Barcelona. Vitrified specimens were prepared by placing 3 µL of a sample on a holey carbon TEM grid, blotted to a thin film and plunged into liquid ethane-N2(l) in the Leica EM GP cryoworkstation. The grids were transferred to a 626 Gatan cryoholder and maintained at −179 °C. The grids were analysed with a Jeol JEM 2011 transmission electron microscope operating at an accelerating voltage of 200 kV. Images were recorded on a Gatan Ultrascan 2000 cooled charge-coupled device (CCD) camera with the Digital Micrograph software package (Gatan).

**Enzymatic treatments.** The protocol followed for heparan sulfate enzymatic digestion experiments was described in Christianson et al.[14]. Briefly, cells were cultured in digestion buffer (DMEM supplemented with 0.5% [wt vol$^{-1}$] BSA and

20 mM Hepes-HCl, pH 7.4) and either untreated or treated with 1.2 mIU mL$^{-1}$ heparinase I and 0.6 mIU mL$^{-1}$ heparinase III (Sigma-Aldrich) for 3 h at 37 °C. Then, EVs were added and incubated for 3 h, followed by flow cytometry analysis (as described later).

**EV labelling**. For EV-uptake experiments, purified EVs were fluorescently labelled using PKH26 Red Fluorescent Cell Linker Midi Kit (Sigma-Aldrich). EVs isolated by ultracentrifugation and resuspended in 1 mL of PBS were labelled with the PKH26 dye (2 μM final concentration) according to the manufacturer's instructions.

**EV purification**. For all experiments, EVs were isolated by ultracentrifugation. Cells were cultured in media supplemented with 10% exosome-depleted FBS and after 48 h were changed to media supplemented with 0.5% exosome-depleted FBS. Equivalent amounts of supernatant fractions collected after 48–72 h cell culture were pelleted by centrifugation at 500 g for 10 min, to remove cells. The supernatant was centrifuged at 12,000 g for 20 min, to remove cell debris and dead cells, and finally, EVs were collected by centrifugation at 100,000 g for 70 min (Thermo Scientific, mx ultra-series centrifuge). The EV pellet was resuspended in 20 mL of phosphate-buffered saline (PBS) and collected by ultracentrifugation at 100,000 g for 70 min (Thermo Scientific, mx ultra-series centrifuge), and the final pellet was resuspended in 100 μL of PBS. Protein concentration in the final EV pellet was measured by BCA (Pierce, Thermo Fisher Scientific) according to the manufacturer's instructions. A detailed protocol for EV preparation is available at Protocol Exchange (Nature Research, https://doi.org/10.21203/rs.3.pex-1044/v1).

**EV gradient separation**. A discontinuous iodixanol gradient was prepared by diluting a stock solution of OptiPrep™ (60% w/v) with 0.25 M sucrose/10 mM Tris, pH 7.5, to generate 40, 20, 10, and 5% w/v iodixanol solutions. The discontinuous iodixanol gradient was generated by sequentially layering 3 mL each of 40, 20, and 10% (w/v) iodixanol solutions, followed by 2.5 mL of the 5% iodixanol solution. EVs, previously purified by ultracentrifugation, were loaded on the discontinuous iodixanol gradient and centrifuged for 16 h at 100,000 g at 4 °C. Fractions of 1 mL were collected from the top of the gradient, and recentrifuged at 100,000 g for 2 h at 4 °C. The resulting pellets were resuspended in 25 μL Laemmli buffer 1× and analysed by Western blot.

**FAK activity**. Starved cells were treated with Dyngo-4a or Pitstop-2 30 min prior to adding EVs; FAK 14 inhibitor was added the night before the EVs. Acceptor cells were incubated with EVs (10–25 μg mL$^{-1}$) for 1 h, then washed with cold PBS and collected on ice for immunoblotting.

**Flow cytometry**. For EV uptake studies, acceptor cells were incubated with PKH-labelled EVs (1–5 μg mL$^{-1}$) for 1–6 h. Cells were detached using enzyme-free PBS-based Cell Dissociation Buffer (Gibco). Cells were washed in PBS solution, suspended in FACS buffer (EDTA 2.5 mM, 1% BSA in PBS), and analysed on a FACSCalibur instrument with integrated FACSDiva (BD Biosciences) v8.0.1. FCS Express 4 Flow Research was used to perform Flow cytometry analysis. In blocking experiments, heparin, Dyngo-4a or Pitstop-2 were added 30 min prior to EVs; FAK 14 inhibitor was added the night before the EVs.

**Homing of MDA.MB.231 cells to lung**. Female athymic nude mice (Harlan Interfauna Iberica, Barcelona, Spain) were kept in pathogen-free conditions and used at 7 weeks of age. Temperature and relative humidity is continuously recorded in the animal housing. The temperature ranged from 23 to 25 °C. Relative humidity ranged between 45 and 65%. Lighting was artificial, from an automatic controlled supply. The cycle gradually simulates twilight and sunset from 7:30 to 8:00 a.m. and from 19:30 to 20:00 p.m., respectively, giving 12 h of light with an intensity of 300 Lux and 12 h of darkness for each 24 h period. Animal care was handled in accordance with the Guide for the Care and Use of Laboratory Animals of the Vall d'Hebron University Hospital Animal Facility, and the experimental procedures were approved by the Animal Experimentation Ethical Committee of the institution (76/17 CEEA).

For the homing assays, we used shCON and shITGB3 MDA.MB.231 cells that were labelled with DiIC18 (D7757; Molecular Probes) according to the manufacturer instructions; 100,000 cells were injected intravenously into the left caudal tail vein of each mouse. Mice were sacrificed 12 h later for lung harvest. The whole lungs were minced with scalpels and incubated in 2 mL of freshly prepared DMEM containing 0.8 mg mL$^{-1}$ dispase (Gibco) and 1.5 mg mL$^{-1}$ collagenase P (Roche) at 37 °C in agitation for 15 min. After incubation, 10 mL of DMEM was added, and tissue fragments and cell suspensions were spun at 500 g for 5 min. Pellets were incubated with 0.05% trypsin-EDTA at 37 °C for 2 min. Single-cell suspensions were filtered through 70-μm nylon filter and washed in PBS with 2 mM EDTA and 1% BSA. Cells were blocked and stained with CD45 Monoclonal Antibody (30-F11) (1:200), FITC (eBioscience) for flow cytometry, and absolute counts of CD45-negative DiIC18-positive cells were obtained and analysed on a FACSCalibur instrument with integrated FACSDiva (BD Biosciences) v8.0.1.

**Immunofluorescence**. Cells were grown on cover glasses for 1 day, rinsed with TBS at room temperature and fixed for 10 min with 4% paraformaldehyde. After rinsing with TBS, in the case of permeabilized conditions, cells were incubated with 0.1% Triton X-100 in TBS for 30 min followed by 30 min in blocking buffer (BSA 2% in TBS-T 0.1% Triton X-100) at room temperature. For non-permeabilized conditions, cells were directly incubated in blocking buffer (BSA 2% in TBS) for 1 h at room temperature. Next, cells were incubated in the suitable primary antibodies for 1 h at room temperature and rinsed repeatedly with TBS before incubating in the appropriate fluorescein-labelled secondary antibody for 1 h at room temperature. Cells were then washed extensively with TBS and mounted on a glass slide in Prolong Diamond Antifade mountant with DAPI (Pierce, Thermo Fisher Scientific). The following primary antibodies were used: anti-EEA1 (1:1000) (Cell Signalling), anti-LAMP-1 (1:1000) (Santa Cruz), anti-avb3 (1:50) clone LM609 (Merck), and anti-tubulin (1:1000) (DM1a-FITC, Sigma). The following secondary antibodies were used: goat anti-mouse IgG (H + L) Alexa Fluor Plus 488 and goat anti-rabbit IgG (H + L) Alexa Fluor 633 (Thermo Fisher Scientific) (1:1000).

**Microscopy and image analysis**. Cells were observed under a confocal laser scanning microscope LSM980 (Carl Zeiss, Germany) with a 63× 1.4 NA oil immersion lens. Images for EV location analysis were captured with a pixel size of 0.07 × 0.07 × 0.31 px μm$^{-1}$ (xyz, respectively). Images for co-localization analysis were optimally sampled (Nyquist–Shannon theorem) at 0.03 × 0.03 × 0.16 px μm$^{-1}$ (xyz, respectively). Images were firstly deconvoluted using Zeiss Zen 3.0 program. EV location and counting was done using Imaris software (7.2.3) though 3D cell reconstruction. Co-localization was done using Zeiss Zen 3.0 program.

**Modulation of expression using plasmid transient transfection**. Wild-pGFP FAK was a gift from Kenneth Yamada (Addgene plasmid # 50515; http://n2t.net/addgene:50515; RRID:Addgene_50515)[65]. Transfections using TransIT-BrCa (Mirrus) were performed according to the manufacturer's protocol, at a final concentration of 3 μg of plasmid DNA in a single well of a 6-well plate. Western blotting with anti-phospho-FAK (Y397) antibody was used to confirm plasmids were expressing the wild type.

**Expression modulation by retroviral and lentiviral infection**. For lentiviral shRNA of ITGB3, pLKO.1-puro-shITGB3 was constructed by annealing the oligonucleotides 5′-CCGGGCCAAGACTCATATAGCATTGCTCGA-3′ and 5′-AATTCAAAAAGCCAAGACTCATATAGCATT-3′ and cloning them into a pLKO1 vector. The shITGB3 #3 was obtained from Dharmacon. We also used pLKO1 as shCTL. Sigma Mission shRNA clone TRCN0000006649 plasmid was used for targeting DYNAMIN 2. WT DYNAMIN 2 pEGFP (Addgene plasmid # 34686; http://n2t.net/addgene:34686; RRID:Addgene_34686) and K44A DYNAMIN 2 pEGFP (Addgene plasmid # 34687; http://n2t.net/addgene:34687; RRID: Addgene_34687), a gift from Sandra Schmid, were used for subcloning (HindIII/NotI) into pLPCX retroviral plasmid. Cell monolayers were incubated with virus containing cell culture supernatant in the presence of 4 μg/ml polybrene (Sigma-Aldrich) for 24 h. Infected MDA.MB.231 and MCF7 cells were selected with 0.7 or 1.5 μg mL$^{-1}$ puromycin, respectively, for 3–4 days. Viral production and infection were performed at 37 °C. All plasmids were sequenced twice from both ends to ensure expression of the correct coding sequence.

**Expression modulation by small interfering RNA knockdown**. siRNA transfections were performed in Opti-MEM medium (Life Technologies) using Lipofectamine RNA-iMAX (Life Technologies), following manufacturer's instructions. Unless otherwise indicated, transfections were performed for 48 h, with 50 nM siRNA. The following siRNAs were used: non-silencing control siRNA (D-001810-01-05) from Dharmacon and siFAK from Sigma-Aldrich (SASI_Hs01_00035697).

**Nanoparticle tracking analysis (NTA)**. EVs collected by ultracentrifugation were analysed at similar dilutions in a Nanosight NS-300 instrument (Malvern Instruments) for real-time characterization of the vesicles. Using the Automatic Syringe Pump (Malvern) at a pump speed of 40, three videos of 60 s were recorded for each sample at 20 °C and a concentration of 10–55 particles per frame and used to calculate mean particle concentration and size. The measurements were analysed using NTA version 3.1 Build 3.1.45 was used.

All NTA and CryoEM studies were performed at the CIBER-BBN (Bioingenieria, Biomateriales y Nanomedicina), Institut de Ciència de Materials de Barcelona (ICMAB-CSIC, Barcelona, Spain) (http://www.nanbiosis.es/u6-e04-nanosight-lm-20-for-nanoparticle-tracking-analysis/).

**Preparation of EVs and cell lysates**. EV pellets prepared by differential high-speed centrifugation were resuspended in PBS and a known concentration was loaded on SDS–polyacrylamide gels in Laemmli buffer. The corresponding cell layers were washed in cold PBS and scraped on ice in lysis buffer (50 mM Tris-HCl, pH 7.4, 150 mM NaCl, 1% Triton X-100, 1% sodium deoxycholate, 0.1% SDS, 1 mM EDTA) supplemented with PhosSTOP and Complete Phosphatase/Protease Inhibitor Cocktails (Roche Diagnostics GmbH, Mannheim, Germany). The whole

cellular lysate was centrifuged at 16,000 g for 15 min to clear cell debris then stored at −20 °C. Protein concentration in the cellular extracts was determined using the BCA method (Pierce, Thermo Fisher Scientific) according to the manufacturer's instructions. Protein extracts (40–50 μg per sample) were loaded onto sodium dodecyl sulfate-polyacrylamide gel electrophoresis gels and transferred electrophoretically to PVDF membranes, and immunodetection of proteins was performed using ECL™ Western Blotting Detection Reagents (GE Healthcare, Buckinghamshire, UK). The following primary antibodies were used: anti-CD81 (1:5000), anti-pFAK (1:1000) and anti-FAK (1:1000), anti-DYNAMIN 2 (1:500), anti-LAMP1 (1:500) and anti-ERK (1:1000) (Santa Cruz Biotechnology); anti-Vinculin (1:1000) (Sigma-Aldrich) anti-TSG101 (1:2500), anti-αv integrin (1:2000) and anti-β1 (1:5000) integrin (Abcam); anti-β3 integrin (1:500), anti-pERK (1:1000), anti-pSrc (1:500), anti-Src (1:500), anti-pAKT (1:1000), anti-AKT (1:1000), anti-EEA1 (1:2500) and anti-EIF4E (1:1000) (Cell Signalling); anti-flotillin-1 (1:1500) (Novus Biologicals) and anti-β-actin (1:500; Sigma-Aldrich). Anti-mouse and anti-rabbit HRP secondary antibodies were from Pierce (1:10000). Bound antibodies were visualized with an enhanced chemiluminescence detection kit (Amersham Pharma-Biotech).

**Proteomics sample preparation**. Eluted proteins were reduced, alkylated and digested to peptide mixes according to the filter-aided sample preparation[66] method using LysC 1:10 ratio (w:w; enzyme:substrate) at 37 °C overnight, followed by trypsin 1:10 ratio (w:w; enzyme:substrate) at 37 °C for 8 h. Tryptic peptide mixtures were desalted using a C18 UltraMicroSpin column[67]. Protein extracts (15 μg) were reduced with dithiothreitol (30 nmol, 37 °C, 60 min) and alkylated in the dark with iodoacetamide (60 nmol, 25 °C, 30 min). Samples were first diluted to 2 M urea with 200 mM NH$_4$HCO$_3$ for digestion with endoproteinase LysC (1:10 w-w, 37 °C, o/n, Wako), and then diluted 2-fold with 200 mM NH$_4$HCO$_3$ for trypsin digestion (1:10 w-w, 37 °C, 8 h, Promega). After digestion, peptide mix was acidified with formic acid and desalted with a MicroSpin C18 column (The Nest Group, Inc) prior to LC–MS/MS analysis.

**Chromatographic and mass spectrometric analysis**. Samples were analysed using an LTQ-Orbitrap Velos Pro mass spectrometer (Thermo Fisher Scientific, San Jose, CA, USA) coupled to an EASY-nLC 1000 (Thermo Fisher Scientific [Proxeon], Odense, Denmark). Peptides were loaded onto the 2-cm Nano Trap column with an inner diameter of 100 μm packed with C18 particles of 5 μm particle size (Thermo Fisher Scientific) and were separated by reversed-phase chromatography using a 25-cm column with an inner diameter of 75 μm, packed with 1.9 μm C18 particles (Nikkyo Technos Co., Ltd. Japan). Chromatographic gradients started at 93% buffer A and 7% buffer B with a flow rate of 250 nL min$^{-1}$ for 5 min and were gradually adjusted to 65% buffer A and 35% buffer B over 120 min. After each analysis, the column was washed for 15 min with 10% buffer A and 90% buffer B. Buffer A: 0.1% formic acid in water. Buffer B: 0.1% formic acid in acetonitrile. The mass spectrometer was operated in positive ionization mode with nanospray voltage set at 2.1 kV and source temperature at 300 °C. Ultramark 1621 was used for external calibration of the FT mass analyser before the analyses, and an internal calibration was performed using the background polysiloxane ion signal at m/z 445.1200. The acquisition was performed in data-dependent acquisition (DDA) mode, and full MS scans with 1 micro scans at a resolution of 60,000 were used over a mass range of m/z 350–2000 with detection in the Orbitrap. Auto gain control (AGC) was set to 1E6, dynamic exclusion (60 s), and charge state filtering disqualifying singly charged peptides was activated. In each cycle of DDA analysis, following each survey scan, the 20 most intense ions with multiple charged ions above a threshold ion count of 5000 were selected for fragmentation. Fragment ion spectra were produced via collision-induced dissociation (CID) at normalized collision energy of 35% and acquired in the ion trap mass analyser. AGC was set to 1E4, with an isolation window of 2.0 m/z, an activation time of 10 ms and a maximum injection time of 100 ms. All data were acquired with Xcalibur software v2.2. Digested bovine serum albumin (New England biolabs cat # P8108S) was analysed between each sample to avoid sample carryover and to assure stability of the instrument, and QCloud[66] was used to control instrument longitudinal performance during the project.

**Data analysis**. Acquired spectra were analysed using the Proteome Discoverer software suite (v1.4, Thermo Fisher Scientific) and the Mascot search engine (v2.5, Matrix Science)[67]. The data were searched against a Swiss-Prot human database plus a list[68] of common contaminants and all the corresponding decoy entries. For peptide identification a precursor ion mass tolerance of 7 ppm was used for MS1 level; trypsin was chosen as the enzyme and up to three missed cleavages were allowed. The fragment ion mass tolerance was set to 0.5 Da for MS2 spectra. Oxidation of methionine and N-terminal protein acetylation were used as variable modifications whereas carbamidomethylation on cysteines was set as a fixed modification. False discovery rate in peptide identification was set to a maximum of 5%. Protein abundance was estimated using the area under the chromatographic peak of the three most intense peptides per protein. Data were log transformed and normalized by equalizing the median of the total protein abundance per sample. Fold changes, p values and q-values were calculated to assess protein relative

quantification. Proteins observed in all replicates of one condition and in none of the other conditions were considered significantly different (presence/absence).

**Statistics**. Results are expressed as means + standard errors of the means. Two-tailed Student's t test and two-way ANOVA multiple comparisons were carried out as appropriate, both using GraphPad Prism 6.01. p < 0.05 was considered significant.

**Reporting summary**. Further information on research design is available in the Nature Research Reporting Summary linked to this article.

## Data availability
The mass spectrometry raw data have been deposited into the PRIDE[69] repository with the dataset identifier PXD013489. All the data supporting the findings of this study are available within the article and its supplementary information files and from the corresponding author upon reasonable request. A reporting summary for this article is available as a Supplementary Information file. Source data are provided with this paper.

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

## Acknowledgements

The authors thank Hector Peinado and his laboratory team (Microenvironment & Metastasis Group, Molecular Oncology Program, Spanish National Cancer Research Centre (CNIO)). The authors thank Dr. Manel Bosch for helpful discussion and assistance with confocal microscopy. The authors thank Dr. Anne Spang and Dr Jordi Berenguer for helpful suggestions and discussion on the paper, and Jane Marshall for language review and editing support. This work was supported by Fondo de Investigaciones Sanitarias (PI14/01320), Redes Temáticas de Investigación Cooperativa en Salud (RD12/0036/0057) and CIBERONC (CB16/12/00363). SRC acknowledges support from the Generalitat de Catalunya (2014 SGR 1131). The CRG/UPF Proteomics Unit is part of the Spanish Infrastructure for Omics Technologies (ICTS OmicsTech) and it is a member of the ProteoRed PRB3 consortium which is supported by grant PT17/0019 of the PE I + D + i 2013-2016 from the Instituto de Salud Carlos III (ISCIII) and ERDF.

## Author contributions

Concept and design: P. Fuentes, S. Hümmer, M. Sesé, S. Ramón y Cajal. Development of methodology: P. Fuentes, S. Hümmer, M. Sesé, H. Peinado. Data acquisition: P. Fuentes, P.J. Guijarro, M. Emperador, S. Sánchez-Redondo H. Peinado. Data analysis and interpretation: P. Fuentes, S. Hümmer, M. Sesé, H. Peinado. Writing: P. Fuentes, S. Hümmer. Review of paper: P. Fuentes, S. Hümmer, M. Sesé, H. Peinado, S. Ramón y Cajal. Study supervision: S. Ramón y Cajal, M. Sesé, S. Hümmer.

## Competing interests

The authors declare no competing interests.
