## [Peer Review File · Nature Communications]

Reviewers' comments:

Reviewer #1 (Remarks to the Author):

This manuscript from the yCajal lab describes experiments aimed at determining a role for α v β 3 integrin in internalisation of extracellular vesicles (EVs). The authors have found that knocking down β 3 integrin reduced the level of EV uptake by MDA-MB-231 cells. They show that activation of focal adhesion kinase (FAK) and dynamin are also required for EV uptake. Inhibition of this EV uptake process seems to reduce the clonal growth of MDA-MB-231 cells, possibly because this renders the cells less sensitive to growth promoting factors in the secretome.

The authors have identified an interesting phenomenon, and the findings would be of interest to the readership of Nat. Comms. However, as the paper stands, the data do not support the conclusions drawn by the authors. Moreover, the paper needs substantially more mechanistic insight into β 3 integrin –dependent EV internalisation before acceptance in Nat. Comms. is justified.

Major points:

1. The FACS-based assay to measure uptake of fluorescently labelled EVs does not discriminate between EVs that are attached to the cell surface and those that are internalised into the cell. High-resolution confocal microscopy to directly visualise labelled EVs inside cells/endosomes would go a long way to support their findings. The authors should also visualise EV internalisation in cells where dynamin and ITGB3 have been inhibited/knocked down to determine whether these factors are indeed required for EV internalisation and not simply binding of EVs to the plasma membrane.
2. To ensure that the defects in EV internalisation observed in Fig 2 are not simply due to reduced levels of ITGB3 at the cell surface, the surface expression of ITGB3 following treatment with FAK14, Dyngo, PitSTOP and Heparin should be measured. Moreover, as FAK14 treatment occurs over a long time-period, the total levels of β 3 integrin should also be measured for this treatment.
3. The small molecule inhibitor of dynamin Dyngo4 has reported off-target activities that inhibit fluid-phase endocytosis. There is a danger that the results in Fig2B and Fig5A linking EV internalisation/FAK activation to dynamin, are artefactual. Therefore, EV internalisation (Fig2B) and EV-induced FAK phosphorylation (Fig5a) should be repeated with knockdown of one or more dynamin isoforms and/or overexpression of dominant negative dynamin (Dynamin-2K44A). Similarly, experiments using the FAK inhibitor FAK14 should be repeated with FAK knockdown where possible.
4. Generally, there is a lack of data providing mechanistic detail on the role of HSPGs and β 3 integrin during EV internalisation. This study would benefit greatly from a set of experiments using high resolution microscopy to dissect: a) The co-localisation of labelled EVs with β 3 at the cell surface. And the requirement of HSPGs for this; b) The internalisation of EV- β 3 complexes/foci within endosomes, for instance EEA1-positive endosomes; c) The recruitment of FAK to EV- β 3 positive foci, and the phosphorylation-status of FAK within these structure.
5. The authors show that treatment with the FAK inhibitor FAK14 reduces EV internalisation (Fig2C) and conclude that EV uptake requires FAK activity. siRNA-mediated knockdown of FAK should achieve similar results. Moreover, it would be predicted that FAK activation/constitutively-active FAK would rescue EV uptake by shITGB3 cells, is this the case?
6. From the NTA profiles shown in Fig3A it appears that shCONTROL cells release far more EVs than shITGB3 cells, however, when normalised to cell number (Fig3C) the opposite is the case. This suggests that there are far fewer shITGB3 cells in the EV preps and the authors should provide an explanation for this. Do shITGB3 cells grow more slowly than shCONTROL cells? As mentioned above, this should be formally tested as the results of this are crucial for the interpretation of additional experiments, e.g. the colony formation assays in (Fig1) and the IMR90 co-culture in (Supp Fig 4).
7. Throughout this study, western blots are performed on EV lysates without the relevant whole cell lysate samples. This means that all data concerning the constituents of shCONTROL vs shITGB3 EVs are difficult to interpret. Molecular weight markers are absent from all western blot images and should be included.

8. The authors show compelling evidence that ITGB3 promotes CD81/TSG101-containing exosome biogenesis. However, the paper would benefit from further characterisation of the EV populations released by shCONTROL and shITGB3 cells. For instance, sucrose or Iodixanol gradients should be used to separate vesicle populations prior to western blotting.
9. The claim that EV-induced activation of FAK is a driver of exosome biogenesis needs to be further supported. The mechanism for this is unclear. The authors should address this by investigating the role of ITGB3/FAK in regulating; a) the biogenesis of ILVs in the MVB; b) MVB dynamics (localisation, trafficking and exocytosis)
10. The authors show that dynamin activity is required for FAK activation following EV treatment. Are additional components of integrin signalling (Src, Akt, Erk) also activated following EV treatment? Moreover, does FAK signalling play a role in colony formation (Fig1A). For instance, do conditioned medium-treated cells have higher levels of phospho-FAK? Do FAK inhibitors impair colony formation in the presence of conditioned media?
11. In Fig 5B the authors use TCA precipitation of cell culture supernatant to show that the levels of secreted CD81 are reduced following FAK inhibition. From this they conclude that FAK activity is responsible for driving exosome biogenesis and secretion, but these data are somewhat preliminary. It is unclear, for instance, why they have not performed western blots on EVs isolated from FAK14-treated cells. This should also be supported with NTA of EVs isolated in the presence of the FAK14 to determine. If the authors are right, this should lead to similar changes in EV size distribution and concentration as those observed in the shITGB3 cells.
12. Generally, the conclusions outlined in the discussion text and in Fig5C are overstated with only preliminary supporting evidence. For instance, the conclusion that FAK activation following EV internalisation promotes syntenin-ALIX-ESCRT recruitment on MVBs is totally unsupported. Either the evidence should be provided, or the text should be toned down.

Minor points

1. The authors show compelling evidence that the EV contents of conditioned medium contribute towards colony formation of MDA-MB-231 cells, and this is subsequently attributed to their uptake of extracellular vesicles, likely exosomes. It is unclear whether large microvesicles/ectosomes or small EVs within the conditioned media are responsible for this however. This could be addressed by assessing the colony forming capacity of CM from which only microvesicles/ectosomes have been depleted, for instance, by subjecting the conditioned media to a 10,000 x g spin.
2. In Fig 2A the authors maintain target cells at 4 °C during incubation with EVs. The aim of this experiment is to assess whether EV internalisation is an energy dependent or a passive process. The design of this experiment is problematic as passive fusion of membrane compartments is likely to be severely inhibited at 4 °C due to reduced membrane fluidity. Can the authors show that passive fusion of membranes is uninhibited at 4 °C?
3. Fibronectin is a described EV cargo and fibronectin-containing EVs have been shown to support formation of cell adhesions (Sung et al, Nat. Comms. 2015). It is a possibility that the authors may be observing ECM-induced FAK activation following EV treatment. Do the EVs isolated from their cells contain fibronectin or other ECM components? If this is the case, the involvement of these ECM components in their EV-induced FAK activation should be investigated.
4. Line number 159 describes Dyngo-4a as a "small molecule inhibitor that blocks GTPase activity". Reference 46 is incorrect- this reference from 2009 is for the drug Dynole where as Dyngo-4a was developed in 2013. Dyngo-4a reduces GTPase activity of dynamin and is believed to act by reducing its ability to oligomerise. The reference they require is Traffic. 2013 Dec; 14(12): 1272-89. 'Building a better dynasore: the dyngo compounds potently inhibit dynamin and endocytosis' McCluskey A et al.

Reviewer #2 (Remarks to the Author):

In this study, Fuentes et al showed that ITGB3 knock-down inhibits extracellular vesicle-uptake (through dynamin-dependent endocytosis), change the composition and size of vesicles present in the culture media of ITGB3 KD cells, and that ITGB3+ vesicles are required for colony formation of breast cancer cell.

The authors propose that ITGB3 controls both exosome uptake and release through activation of the Focal Adhesion Kinase, and speculate that endosome could be the signalling platform.

The experiments are well designed and executed, and well-presented. The manuscript is clear and well written. However, the statements are very strong and not always supported by the data.

I have two major concerns that need to be addressed (especially the first one):

1)

In Figure 2a and B, the authors use a 4 degrees C temperature block to inhibit energy dependent internalization. To my knowledge, ligand-receptor interaction is not abolished at 4°C. A good example is the binding of LDL to its receptor that still occurs at 4 degrees C, although LDL internalization is abolished (Brown and Goldstein PNAS 1974). Therefore, one would expect to detect fluorescently labelled exosomes at the surface of the acceptor cells at 4 degrees C. This is not the case in fig 2a and b. This suggests to me that ITGB3 does not contribute to exosome docking at the surface of the cell, which contradicts author's statement and conclusion. ITGB3 may be involved somehow in EV endocytosis, but does not seem to be sufficient to enable docking of exosomes to the target cell. This diminishes greatly the finding of the paper.

2)

The authors concluded that ITGB3 controls also exosome release because they found that exosomes emanating from ITGB3KD cell are different in size and composition. Another interpretation is that ITGB3 is dedicated to the internalization of a specific subtype of exosomes. This population would correspond to small EVs (50 -100nm) that would be depleted of ESCRT complex and other hits found in the paper. This would also explain why in fig 2A and B vesicle-internalization is reduced by a factor of 2 (i.e. the larger EV are still internalized). The authors might consider discussing this possibility.

To conclude, the first point is an important caveat that needs to be addressed by the authors, because it changes dramatically the conclusion made by the authors.

Response to Reviewers' comments:

Reviewer #1 (Remarks to the Author):

This manuscript from the y Cajal lab describes experiments aimed at determining a role for avb3 integrin in internalisation of extracellular vesicles (EVs). The authors have found that knocking down b3 integrin reduced the level of EV uptake by MDA-MB-231 cells. They show that activation of focal adhesion kinase (FAK) and dynamin are also required for EV uptake. Inhibition of this EV uptake process seems to reduce the clonal growth of MDA-MB-213 cells, possibly because this renders the cells less sensitive to growth promoting factors in the secretome.

The authors have identified an interesting phenomenon, and the findings would be of interest to the readership of Nat. Comms. However, as the paper stands, the data do not support the conclusions drawn by the authors. Moreover, the paper needs substantially more mechanistic insight into b3 integrin –dependent EV internalisation before acceptance in Nat. Comms. is justified.

We really appreciate that the Reviewer thought our paper interesting and of interest to the readership of Nat. Comms. We agree that greater insight into the mechanism here described would strengthen the manuscript and we have undertaken extensive additional experiments and revision guided by the Reviewers' suggestions.

Major points:

1. The FACS-based assay to measure uptake of fluorescently labelled EVs does not discriminate between EVs that are attached to the cell surface and those that are internalised into the cell. High-resolution confocal microscopy to directly visualise labelled EVs inside cells/endosomes would go a long way to support their findings. The authors should also visualise EV internalisation in cells where dynamin and ITGB3 have been inhibited/knocked down to determine whether these factors are indeed required for EV internalisation and not simply binding of EVs to the plasma membrane.

We agree with the Reviewer's concern that the FACS-based measurements presented in the original version of the manuscript cannot discriminate between surface-bound and internalized extracellular vesicles. To strengthen our data, we have performed additional experiments, which are included in the revised version of the manuscript:

1. We included a citric acid wash step before the measurement by FACS to stringently remove surface-bound extracellular vesicles as described in Du Feng *et al*/2010 (new reference 44 in the new version of the manuscript).
2. We determined the extracellular vesicle uptake by confocal microscopy and 3D reconstruction of cells to discriminate between surface-bound and internalized extracellular vesicles.

The results of both approaches confirmed our initial observation that extracellular vesicle uptake is dependent on ITGB3 and dynamin. The relative abundance of vesicles on the cell surface is not significantly altered in shITGB3 cells or after inhibition of dynamin compared to the control cells. Furthermore, these data indicate that the interaction of EVs with the cell surface is transient and only results in vesicle uptake if the initial recognition is linked to ITGB3-dependent activation of endocytosis. These new data are included in new Figure 2a and new Figure 3.

Moreover, we performed confocal microscopy-based co-localization studies of fluorescently-labelled EVs with proteins involved in intracellular vesicle trafficking such as EEA1 for early endosomes and Lamp1 for late endosomes/lysosomes. (see reply to comment 4).

2. To ensure that the defects in EV internalisation observed in Fig 2 are not simply due to reduced levels of ITGB3 at the cell surface, the surface expression of ITGB3 following treatment with FAK14,

Dyngo, PitSTOP and Heparin should be measured. Moreover, as FAK14 treatment occurs over a long time-period, the total levels of b3 integrin should also be measured for this treatment.

We understand the concern of the Reviewer. To exclude the possibility of reduced ITGB3 levels at the cell surface after the indicated treatments, we measured the cell surface abundance of ITGB3 by FACS. As shown in Supplementary Figure 8a, none of the treatments reduced the levels of ITGB3 on the cell surface.

However, inhibition of dynamin by Dyngo4a resulted in elevated levels of ITGB3 on the cell surface. These results are in line with previous reports describing the dynamin-dependent process of integrin trafficking as a key mechanism to regulate its function at the cell surface. Moreover, the results highlight that not solely the presence of integrin beta 3 on the cell surface, but the active recycling of the integrin in a dynamin-dependent manner appears to be crucial for the uptake of EVs.

Furthermore, the treatment with the FAK14 inhibitor revealed that ITGB3 protein levels are not affected. These new data are included as the Supplementary Figure 8.

3. The small molecule inhibitor of dynamin Dyngo4 has reported off-target activities that inhibit fluid-phase endocytosis. There is a danger that the results in Fig2B and Fig5A linking EV internalisation/FAK activation to dynamin, are artefactual. Therefore, EV internalisation (Fig2B) and EV-induced FAK phosphorylation (Fig5a) should be repeated with knockdown of one or more dynamin isoforms and/or overexpression of dominant negative dynamin (Dynamin-2K44A). Similarly, experiments using the FAK inhibitor FAK14 should be repeated with FAK knockdown where possible.

We are thankful for the notion that the small molecule inhibitor of Dyngo has a reported off-target activity in fluid phase endocytosis. To exclude that this off-target effect is the underlying cause of our results, we have repeated the experiments with knockdown of dynamin-2 by shRNA and overexpressing dominant negative dynamin-2 (dynamin-2K44A). As shown in Fig. 2b, Fig. 2c, Fig. 3c and Fig.3d, knockdown of Dyn2 by shRNA or overexpression of the dominant negative version of dynamin-2 (dynamin-2K44A) confirm the results obtained with the dynamin inhibitor Dyngo.

Regarding the FAK14 inhibitor, we have performed a knockdown of FAK as suggested by the Reviewer. We have been able to recapitulate the phenotype of the FAK14 inhibitor in extracellular vesicle uptake using siRNA-mediated knockdown of FAK. These new data are included as a Fig. 6b.

4. Generally, there is a lack of data providing mechanistic detail on the role of HSPGs and b3 integrin during EV internalisation. This study would benefit greatly from a set of experiments using high resolution microscopy to dissect: a) The co-localisation of labelled EVs with b3 at the cell surface. And the requirement of HSPGs for this; b) The internalisation of EV-b3 complexes/foci within endosomes, for instance EEA1-positive endosomes; c) The recruitment of FAK to EV-b3 positive foci, and the phosphorylation-status of FAK within these structure.

We considered the Reviewer's advice and agree that the study would benefit from a set of experiments using high-resolution microscopy. We have performed the requested experiments with high-resolution confocal microscopy. In the following section, we present the results obtained and discuss the difficulties faced during the consecution of the experiments:

Response for: a) co-localisation of labelled EVs with b3 at the cell surface.

Supporting Figure 1. Cells were treated for 1h with PKH26-labelled EVs (red), fixed, and stained for avb3 (green).

To visualize co-localization of EVs at the cell surface, cells were treated for 1h with PKH26-labelled EVs (red), fixed, and stained under non-permeabilizing conditions for ITGB3 (avb3) (green). As shown in the picture, a clear co-localization between ITGB3 (avb3) and EVs can be observed (white arrows). However, the interpretation of these results is hampered by the fact that ITGB3 is not only a cell surface receptor but also found in exosomes. As we indicate in the picture, extracellular vesicles are also positive for avb3 (yellow arrows). Thus, even though we see a clear co-localization between ITGB3 and EVs, we are not able to discriminate if this reflects the co-localization of cellular ITGB3 with EVs or if we detect the portion of ITGB3 that is present in the EVs.

Response on comment 2) High-resolution confocal microscopy to directly visualise labelled EVs inside cells/endosomes

Cells were treated for 1h with PKH26-labelled EVs, fixed, permeabilized and stained for EEA1 and Lamp1.

As shown in the picture and the graph, PKH26-labelled EVs can readily co-localize with marker proteins of intracellular transport. Furthermore, the co-localization with EEA1 suggests that those vesicles enter into the receiving cell via endocytosis. Notably, after 1h of treatment, the majority of vesicles is found in Lamp1-positive structures, highlighting the quick dynamic process of EV uptake and processing. As shown before by FACS and microscopy-based analysis, EV uptake in shITGB3 cells is severely impaired. The remaining portion of EVs, still entering into the shITGB3 cells show no significant difference in their co-localization with EEA1 and Lamp1.

Supporting Figure 2. Cells were treated for 1h with PKH26-labelled EVs (white), fixed, and stained for EEA1 (red) and Lamp1 (green). Right: Graphical representation of the Mander's coefficient values for determining co-localization between the different markers. (n=25 cells/condition).

Response for b) internalisation of EV-b3 complexes/foci within endosomes, for instance EEA1-positive endosomes

To visualize co-localization of EVs with ITGB3 and EEA1, cells were treated for 1h with PKH26-labelled EVs, fixed, permeabilized and stained for ITGB3 (avb3) and EEA1.

Supporting Figure 3. MDA.MB.231 cells were treated for 1h with PKH26-labelled EVs (white), fixed, and stained for EEA1 (red) and avb3 (green). Right: Graphical representation of the Mander's coefficient values for determining co-localization between the different markers. (n=25cells/condition).

As shown in Supporting Figure 3, a strong co-localization between ITGB3 (avb3) and EVs can be observed. As described above, we are, however, not able to discriminate if this reflects the co-localization of cellular ITGB3 with EVs or if we were detecting ITGB3 that is present in the EVs.

Response for c) The recruitment of FAK to EV-b3 positive foci, and the phosphorylation-status of FAK within these structures.

Cells were treated for 1h with PKH26-labelled EVs, fixed, permeabilized and stained for p-FAK.

Supporting Figure 4 MDA.MB.231 cells were treated for 1h with PKH26 labelled EVs (red), fixed and stained for p-FAK (green)

As shown in the picture, staining of MDA.MB.231 with p-FAK antibody results in a strong staining close to the cell membrane which may be derived from the concentrated activity of the kinase at focal adhesions. This however might mask a cytosolic pool of the kinase and prevented us from performing the co-localization analysis with internalized EVs.

In summary, confocal microscopy has allowed us to confirm the co-localization of EVs with EEA1 and Lamp1. However, technical difficulties have precluded us from precisely discriminating the co-localization of internalized EVs with ITGB3 and pFAK. We would like to point out, however, that the maximal resolution that can be achieved with this technique is approximately 200nm and can therefore not be used to detect single EVs.

Nevertheless, in an attempt to respond to the Reviewer's questions, we used in parallel a biochemical approach. We performed a sucrose step gradient to fraction cellular vesicles by density. In line with previous results (Arriagada C *et al.* 2019, Alanko J *et al.* 2015) describing the role of FAK on early endosomes, the p-FAK enrichment in fraction one coincides with the early endosome marker EEA1. While ITGB3 can be detected across the different fractions, the presence of ITGB3 in fraction one might support that ITGB3 and FAK act together on early endosomes. Interestingly, Src, another interaction partner of ITGB3, appears in later fractions of the gradient, coinciding with the late endosome/lysosome marker Lamp1. Thus, ITGB3 may have different interaction partners within different intracellular vesicle populations. CD81 can be also detected in these fractions.

Supporting Figure 5. Endosomal fractionation

Endosomal fractionation was carried out as described previously (Araújo ME *et al.* Cold Spring Harb Protoc. 2015). In brief, post nuclear supernatant (PNS) was obtained by passing the cell suspension through a 29-gauge needle, followed by 1,800 g centrifugation for 10 minutes. A discontinuous gradient was generated (bottom to top: 45% sucrose (2 mL), 35% sucrose (3.5 mL), 25% sucrose (3.5 mL)) and 2 mL of PNS were layered on top of the gradient. Following centrifugation for 1 hour at 100,000 g, fractions of 1.8 mL were collected from top (F1) to bottom (F6), diluted 1:1 in lysis buffer without sucrose and centrifuged at 100,000 g for 1 hour. The resulting pellets were collected in Laemmli buffer and analyzed by Western blotting. Top: Western blot analysis of PNS and the different endosomal fractions for the indicated proteins. Bottom: Amido Black staining of PVDF membrane.

5. The authors show that treatment with the FAK inhibitor FAK14 reduces EV internalisation (Fig2C) and conclude that EV uptake requires FAK activity. siRNA-mediated knockdown of FAK should achieve similar results. Moreover, it would be predicted that FAK activation/constitutively-active FAK would rescue EV uptake by shITGB3 cells, is this the case?

We agree with the reviewers' comments. As stated in the response to comment 3, knockdown of FAK by siRNA recapitulates the phenotype of FAK14 treatment in vesicle internalization (see new Fig. 6b).

To address the question of whether FAK activation can rescue the defect in EV uptake in shITGB3 cells, we have performed the following experiment: we transiently overexpressed GFP-FAK in shCTL and shITGB3 cells, measured with FACS the vesicle uptake into GFP-positive cells and determined the FAK activity by Western blot. As shown in new Fig. 6d-e, while we can observe a clear increase in FAK activity, the EV uptake in shITGB3 cells is not restored.

This result is in line with our proposed model. As a transmembrane cell surface protein, ITGB3 occupies a critical position at the interface of extracellular and cellular interactions and plays a dual role in the process of EV uptake. The recognition of EVs on the cell surface by ITGB3-interacting HSPGs can be transmitted into the cytoplasm to trigger the interplay between dynamin and FAK, which ultimately allows the endocytic uptake of EVs. Accordingly, activation of FAK can only restore the second step in the process of EV uptake, but is not able to complement the function of ITGB3 in the HSPG-dependent recognition of EVs at the cell surface. These new data are included in Fig. 6b-e and discussed in the new version of the manuscript.

6. From the NTA profiles shown in Fig3A it appears that shCONTROL cells release far more EVs than shITGB3 cells, however, when normalised to cell number (Fig3C) the opposite is the case. This suggests that there are far fewer shITGB3 cells in the EV preps and the authors should provide an explanation for this. Do shITGB3 cells grow more slowly than shCONTROL cells? As mentioned above, this should be formally tested as the results of this are crucial for the interpretation of additional experiments, e.g. the colony formation assays in (Fig1) and the IMR90 co-culture in (Supp Fig 4).

We apologize for the inadequate representation of the results in Figure 3A. We have moved this figure to the supplementary material and replaced it with the new Figure 4A. New Figure 4A now includes the histograms of the number of particles/mL/10⁶cell of control and shITGB3 cells in the same graph. This graph highlights the difference in the abundance of different extracellular vesicle populations in control and shITGB3 cells.

As we stated in the original figure legends, the graphs in Fig. 3c only took into account the number of vesicles defined as exosomes (up to 125nm). This fraction is enriched in the supernatant of shITGB3 cells. In contrast, extracellular vesicles in the size of 120nm to 500nm are reduced in the supernatant of shITGB3 cells. To highlight the differences between the vesicle populations we have added 3 graphical representations, showing the total number of particles/mL/10⁶cell (Fig.4b right panel), the number corresponding to small vesicles (50-125nm) (Fig. 4b left panel) and the number corresponding to larger vesicles (>125nm) (Fig. 4b middle panel) per 10⁶cell in the supernatant of shITGB3 and control cells.

Regarding the question on differences in cell growth, we would like to draw the attention of the reviewer to our previous publication (Sese et al. Oncotarget 2017). In this study we demonstrated that the cell growth of shITGB3 cells is only significantly different under hypoxic conditions. Under normal growth conditions, which were used throughout this work, no significant differences in cell proliferation could be observed.

7. Throughout this study, western blots are performed on EV lysates without the relevant whole cell lysate samples. This means that all data concerning the constituents of shCONTROL vs shITGB3

EVs are difficult to interpret. Molecular weight markers are absent from all western blot images and should be included.

We agree with the Reviewer's comment. We have now replaced the original Fig. 4c with new Fig. 5d, showing total cell lysates and extracellular vesicles on the same membrane. In addition, we have added the corresponding molecular weight marker in all blot images of the manuscript.

8. The authors show compelling evidence that ITGB3 promotes CD81/TSG101-containing exosome biogenesis. However, the paper would benefit from further characterisation of the EV populations released by shCONTROL and shITGB3 cells. For instance, sucrose or Iodixanol gradients should be used to separate vesicle populations prior to western blotting.

We agree with the Reviewer's comment that additional characterization of the EV populations would further support our findings. As suggested, we have performed iodixanol gradients from extracellular vesicles, isolated from shCON and shITGB3 cells and present this data in new Fig. 5 e-f. In the same gel, we also ran the positive fractions for exosomes markers (F6/F7) together with the input of both conditions shCON and shITGB3-derived EVs in order to discard differences due to exposure times. The obtained results confirm that the CD81/TSG101 population of EVs is strongly reduced in the supernatant of shITGB3 cells. Whether this result can be interpreted as altered exosome biogenesis or a selective uptake defect of certain vesicle populations is further discussed in the next comment.

9. The claim that EV-induced activation of FAK is a driver of exosome biogenesis needs to be further supported. The mechanism for this is unclear. The authors should address this by investigating the role of ITGB3/FAK in regulating; a) the biogenesis of ILVs in the MVB; b) MVB dynamics (localisation, trafficking and exocytosis)

The Reviewer's comment is in line with the suggestion of Reviewer 2 (comment 2) that the altered presence of vesicles in shITGB3 or after inhibition of FAK can be seen as the result of either defect in exosome release/biogenesis or exosome subtype-specific effects in uptake into the receiving cell. In our original manuscript we only proposed that FAK might be also involved in EV biogenesis. Nevertheless, in the revised version of the manuscript, this statement has been rephrased to tone down this message. Our manuscript highlights the previously unknown role of ITGB3/FAK in the uptake of extracellular vesicles for which we provide compelling evidence throughout the manuscript. As this appears to be the critical role in clonogenic growth, most relevant for cancer metastasis, the study is based on the role and the interplay of FAK/ITGB3 in vesicle uptake. The observation that interference with the function of both proteins also results in the same alterations in the composition of vesicles in the cell culture supernatant, strengthens the functional link between ITGB3 and FAK in the uptake of the same vesicle population. The possibility that the changes in EV subpopulation in the cell culture supernatant might arise from defects in exosome biogenesis, is only suggested. However, we believe that elucidating the exact molecular mechanisms underlying this particular function is beyond the scope of the presented manuscript.

10. The authors show that dynamin activity is required for FAK activation following EV treatment. Are additional components of integrin signalling (Src, Akt, Erk) also activated following EV treatment? Moreover, does FAK signalling play a role in colony formation (Fig1A). For instance, do conditioned medium-treated cells have higher levels of phospho-FAK? Do FAK inhibitors impair colony formation in the presence of conditioned media?

As suggested by the Reviewer, we have also analyzed the role of other components of the integrin signaling cascade and present the obtained data in new Supplementary Fig. 9. In brief, activity of the other tested components of the integrin signaling pathway appear to be either not affected (Erk (p-p44/p-42 MAPK), AKT (S473)) or even diminished (Src (Y416)) in response to EV treatment. Thus,

among all tested members of the known integrin signaling pathway, only the activity of FAK positively correlates with EV stimulation.

We have also addressed the role of FAK in colony formation. We demonstrate that colony growth is accompanied with an increase in FAK activity (p-FAK) particularly in the presence of conditioned media. FAK activity has furthermore been demonstrated to be required for clonogenic growth (Vita M. Golubovskaya *et al.* 2012). Treatment of cells with the FAK14 inhibited clonogenic growth in the absence and presence of conditioned media (data not shown). Given the multitude of processes that are regulated by FAK, however, it is not surprising that this the activity of FAK is already required in normal growth conditions for clonogenic growth (Johnson, T.R *et al.* 2008; Begum, A *et al.* 2017). This hampers analysis of the role of FAK under conditioned media stimulated conditions.

11. In Fig 5B the authors use TCA precipitation of cell culture supernatant to show that the levels of secreted CD81 are reduced following FAK inhibition. From this they conclude that FAK activity is responsible for driving exosome biogenesis and secretion, but these data are somewhat preliminary. It is unclear, for instance, why they have not performed western blots on EVs isolated from FAK14-treated cells. This should also be supported with NTA of EVs isolated in the presence of the FAK14 to determine. If the authors are right, this should lead to similar changes in EV size distribution and concentration as those observed in the shITGB3 cells.

We agree with the reviewer's comment and to strengthen our initial conclusion, we have repeated the isolation of EVs by ultracentrifugation. We were not able to determine the size distribution of the obtained vesicles, focusing our efforts instead on completing the Western blot data. These data confirmed our previous results, as CD81 was strongly reduced in the EV fraction of FAK14-treated cells. Like in the case of EVs obtained from shITGB3 cells, the reduction of CD81 was accompanied by reduced levels of TSG101. The similarity in the changes in EVs obtained from shITGB3 cells and after inhibition of FAK are furthermore supported by the EV marker proteins flotillin-1 and actin which were unaffected in both cases. These new data are included in Fig. 6f.

12. Generally, the conclusions outlined in the discussion text and in Fig5C are overstated with only preliminary supporting evidence. For instance, the conclusion that FAK activation following EV internalisation promotes syntenin-ALIX-ESCRT recruitment on MVBs is totally unsupported. Either the evidence should be provided, or the text should be toned down.

We apologize for overstating and, following the Reviewer's suggestion, we have toned down the statement on the molecular link between EV internalization and exosome biogenesis. The central message of the manuscript is the role of ITGB3/FAK in extracellular vesicle uptake, crucial for conditioned media-stimulated clonogenic growth, which might link the function of this complex to the described role of both proteins in cancer metastasis.

In line with this, we have modified our initial proposal model as shown in new Fig. 6h, focusing on the HSPGs-ITGB3-FAK EV-dependent uptake.

Minor points

1. The authors show compelling evidence that the EV contents of conditioned medium contribute towards colony formation of MDA-MB-231 cells, and this is subsequently attributed to their uptake of extracellular vesicles, likely exosomes. It is unclear whether large microvesicles/ectosomes or small EVs within the conditioned media are responsible for this however. This could be addressed by assessing the colony forming capacity of CM from which only microvesicles/ectosomes have been depleted, for instance, by subjecting the conditioned media to a 10,000 x g spin.

We agree with the comment of the reviewer that ultracentrifugation-based depletion of extracellular vesicles results in a depletion of small (exosomes) and larger (microvesicles/ectosomes) vesicles. To address whether microvesicles/ectosomes within the conditioned media are required for the increase in the colony forming capacity, we have assessed the colony forming capacity of conditioned media subjected to a 10,000 g spin. The CM and CM subjected to a 10,000 g spin were indistinguishable from each other in their capacity to increase colony formation. These new data are included in new Supplementary Fig. 2.

2. In Fig 2A the authors maintain target cells at 4 oC during incubation with EVs. The aim of this experiment is to assess whether EV internalisation is an energy dependent or a passive process. The design of this experiment is problematic as passive fusion of membrane compartments is likely to be severely inhibited at 4 oC due to reduced membrane fluidity. Can the authors show that passive fusion of membranes is uninhibited at 4 oC?

We thank the Reviewer for the advice. As we provide in the following experiments compelling evidence that EV uptake in an ITGB3-dependent manner is achieved by dynamin-dependent endocytosis, we have moved this initial experiment to supplementary material. We comment on this particular experiment in comment 1 of Reviewer 2.

3. Fibronectin is a described EV cargo and fibronectin-containing EVs have been shown to support formation of cell adhesions (Sung et al, Nat. Comms. 2015). It is a possibility that the authors may be observing ECM-induced FAK activation following EV treatment. Do the EVs isolated from their cells contain fibronectin or other ECM components? If this is the case, the involvement of these ECM components in their EV-induced FAK activation should be investigated.

We agree with the comment of the Reviewer that there is the possibility that ECM proteins might be responsible for the activation of FAK. We have reanalyzed the proteome obtained from EVs of MDA.MB.231 cells and have detected 14 other ECM components in addition to fibronectin (see table below). Given this number and the fact that not a single ECM component but combinations of those might account for the activation of FAK, we believe that the detailed analysis of the EV cargo responsible for FAK activation could be the subject of a separate future study.

GO_EXTRACELLULAR_MATRIX	
Gene Symbol	Gene Description
GPC1	glypican 1 [Source:HGNC Symbol;Acc:HGNC:4449]
FN1	fibronectin 1 [Source:HGNC Symbol;Acc:HGNC:3778]
CSPG4	chondroitin sulfate proteoglycan 4 [Source:HGNC Symbol;Acc:HGNC:2466]
CASK	calcium/calmodulin dependent serine protein kinase [Source:HGNC Symbol;Acc:HGNC:1497]
HSPG2	heparan sulfate proteoglycan 2 [Source:HGNC Symbol;Acc:HGNC:5273]
TNC	tenascin C [Source:HGNC Symbol;Acc:HGNC:5318]
ERBIN	erbb2 interacting protein [Source:HGNC Symbol;Acc:HGNC:15842]
F3	coagulation factor III, tissue factor [Source:HGNC Symbol;Acc:HGNC:3541]
BCAM	basal cell adhesion molecule (Lutheran blood group) [Source:HGNC Symbol;Acc:HGNC:6722]
MFGE8	milk fat globule-EGF factor 8 protein [Source:HGNC Symbol;Acc:HGNC:7036]
SDC2	syndecan 2 [Source:HGNC Symbol;Acc:HGNC:10659]
GPC6	glypican 6 [Source:HGNC Symbol;Acc:HGNC:4454]
COL5A1	collagen type V alpha 1 chain [Source:HGNC Symbol;Acc:HGNC:2209]
COL18A1	collagen type XVIII alpha 1 chain [Source:HGNC Symbol;Acc:HGNC:2195]
LAMB3	laminin subunit beta 3 [Source:HGNC Symbol;Acc:HGNC:6490]

Supporting Table 1: Proteins of the extracellular matrix identified by mass-spectrometry of MDA.MB.231 derived EVs.

4. Line number 159 describes Dyngo-4a as a “small molecule inhibitor that blocks GTPase activity”. Reference 46 is incorrect- this reference from 2009 is for the drug Dynole where as Dyngo-4a was

developed in 2013. Dyngo-4a reduces GTPase activity of dynamin and is believed to act by reducing its ability to oligomerise. The reference they require is Traffic. 2013 Dec;14(12):1272-89. 'Building a better dynasore: the dyngo compounds potently inhibit dynamin and endocytosis' McCluskey A et al.

We apologize for this mistake. We have corrected the reference as suggested by the Reviewer.

Reviewer #2 (Remarks to the Author):

In this study, Fuentes et al showed that ITGB3 knock-down inhibits extracellular vesicle- uptake (through dynamin-dependent endocytosis), change the composition and size of vesicles present in the culture media of ITGB3 KD cells, and that ITGB3+ vesicles are required for colony formation of breast cancer cell.

The authors propose that ITGB3 controls both exosome uptake and release through activation of the Focal Adhesion Kinase, and speculate that endosome could be the signalling platform.

The experiments are well designed and executed, and well-presented. The manuscript is clear and well written. However, the statements are very strong and not always supported by the data.

I have two major concerns that needs to be addressed (especially the first one):
1)

In Figure 2a and B, the authors use a 4 degrees C temperature block to inhibits energy dependent internalization. To my knowledge, ligand-receptor interaction is not abolished at 4°C. a good example is the binding of LDL to its receptor that still occurs at 4 degrees C, although LDL internalization is abolished (Brown and Goldstein PNAS 1974). Therefore, one would expect to detect fluorescently labelled exosomes at the surface of the acceptor cells at 4 degrees C. This is not the case in fig2 a and b. This suggests to me that ITGB3 does not contribute to exosome docking at the surface of the cell, which contradicts author's statement and conclusion. ITGB3 may be involved somehow in EV endocytosis, but does not seem to be sufficient to enable docking of exosomes to the target cell. This diminishes greatly the finding of the paper.

We thank the Reviewer for the comment and the opportunity to explain our findings and their interpretation in more detail.

Firstly, we would like to draw the attention of the Reviewer to two other previous manuscripts where complete inhibition of vesicle uptake at 4°C has been reported. (Schneider DJ *et al* JBC 2017 and Emam SE *et al* Scientific Reports 2018).

Regarding the interpretation of the experiment presented in Figure 2a and b, we would like to highlight that at 4 °C the amount of vesicles detected by FACS is strongly reduced and indistinguishable in WT and shITGB3 cells. In particular, the fact that at 4°C no EVs are detected in WT cells rather indicates that none of the putative mechanisms for exosome recognition and uptake is operating under these conditions, but does not exclude ITGB3 as an important player in this process.

We would also like to point out that in our proposed model, ITGB3 does not act as a “receptor” *per se*, but rather as a molecular linker between the HSPG-dependent recognition on the cell surface and the initiation of dynamin-driven endocytosis. In contrast to the LDL/receptor model, our experiments indicate that recognition and/or internalization of vesicles relies on a different mechanism than the classical LDL/receptor interaction. We provide compelling evidence throughout the manuscript that the initial docking of vesicles on the surface of the cell relies on HSPGs (Heparin treatment (Fig. 2e and Fig. 3a) and Heparinase treatment (Fig. 2f). Notably, our observations, regarding vesicle docking by HSPGs and a complete reduction of vesicle uptake at 4°C are in agreement with the work by Christianson HC *et al* (PNAS 2013).

Furthermore, we have now assessed the presence of EVs on the cell surface under different experimental conditions by two independent approaches:

1) We included a citric acid wash step before the measurement by FACS to stringently remove surface-bound extracellular vesicles as described in Du Feng *et al* 2010, reference number 44 in the new version of the manuscript.

2) We determined the extracellular vesicle uptake by confocal microscopy and 3D reconstruction of cells to discriminate between surface-bound and internalized extracellular vesicles.

The obtained data indicate that, even after inhibition of dynamin, none of the tested experimental conditions showed a significant accumulation of EVs at the cell surface. We present the results of these experiments in new Fig. 2a and Fig. 3. Together, these data indicate that the interaction of EVs with the cell surface is a highly dynamic process and if attachment of EVs is not intimately linked to endocytosis, EVs might eventually detach again from the cell surface.

Internalization of EVs, however, requires the concerted activity of vesicle recognition by HSPGs and ITGB3. We propose in our revised model (new Fig. 6h) that the known interaction between HSPGs and ITGB3 allows EV recognition via HSPGs on the cell surface. ITGB3, as a transmembrane cell surface protein, occupies a critical position at the interface of extracellular and cellular interactions. In this way, the recognition of EVs on the cell surface can be transmitted into the cytoplasm to trigger the interplay between dynamin and FAK, which ultimately allows the endocytic uptake of EVs.

Finally, the experiment presented in original Figure 2a and B was thought to demonstrate the energy dependence of extracellular vesicle uptake. As Reviewer 1 points out (minor point 2), this interpretation might be hampered by the fact that membrane fluidity as well as energy-dependent processes might be severely affected at 4°C. We now provide direct evidence in the experiments performed that ITGB3-dependent vesicle uptake is a dynamin- and therefore energy-dependent process, we have moved the experiment of vesicle uptake at 4°C to the supplementary material (Supplementary Fig. 4).

2) The authors concluded that ITGB3 controls also exosome release because they found that exosomes emanating from ITGB3KD cell are different in size and composition. Another interpretation is that ITGB3 is dedicated to the internalization of a specific subtype of exosomes. This population would correspond to small EVs (50 -100nm) that would be depleted of escrt complex and other hits found in the paper. This would also explain why in fig2A and B vesicle-internalization is reduced by a factor r^2 (i.e the larger EV are still internalized). The authors might consider discussing this possibility.

We highly appreciate the comment of the Reviewer, because it is an excellent point, to which we have now devoted further discussion. As the reviewer points out, alterations in the composition of EVs in the cell culture medium of shITGB3 cells might be caused by a defect in exosome biogenesis or be a consequence of selective vesicle uptake. Our manuscript is dedicated to the thus far unknown role of ITGB3/FAK in the uptake of extracellular vesicles. In light of this, we revised the conclusions drawn from the experiments, in which we describe the alterations in the composition of EVs in the cell culture supernatant of shITGB3 cells. In the revised version of the manuscript, we present these results as a possible consequence of the selective defect in EVs uptake. However, as we cannot formally discriminate between defects in exosome biogenesis and EV uptake, we also discuss both possibilities in the revised version of the manuscript.

Reviewers' comments:

Reviewer #1 (Remarks to the Author):

This revision has significantly improved the paper and I appreciate the authors taking my suggestions on board. I feel, however, that some issues remain as the paper stands, particularly regarding the visualisation of internalised EVs in recipient cells by confocal microscopy.

Main points:

Imaging –

1.

a) In order to overcome difficulties in distinguishing between donor and recipient ITGB3-positive structures using confocal microscopy (Supporting Figure 1), the authors could repeat these imaging experiments using EVs derived from ITGB3-knockdown cells. This could work as loss of ITGB3 in donor cells did not affect recipient cell-EV uptake (Figure 1). The experiment displayed in rebuttal letter Supporting Figure 3 could also be repeated using EVs derived from ITGB3-knockdown cells.

b) In supporting Figure 4, the authors were not able to demonstrate the colocalisation between labelled EVs and p-FAK in recipient cells. Although we agree with the authors that confocal microscopy is at the limit of resolution for detection of single EVs, it is well established that EVs accumulate within the endosomal system upon internalisation in recipient cells, and EV-positive vesicular structures can be usually clearly visualised using such microscopy techniques. Are the PKH26-positive structures associated with focal adhesions? Live cell imaging of recipient cells transiently expressing GFP-FAK +/- ITGB3si in the presence of labelled EVs would help characterising the temporal events of FAK activation downstream of EV uptake, and to answer whether EVs colocalise with FAK in recipient cells.

c) In the rebuttal letter the authors include images of labelled EVs with ITGB3 and these images look a lot better than the ones shown in figure 3d. I think the other points made in the rebuttal letter should also be included in the paper. Also fig 3d would benefit from showing each channel separately black on white background as this would be much clearer.

2. The authors show that depletion of EVs from CM impairs its ability to promote clonal growth in MDA-MB-231 cells (Fig. 1). However, instead of experimentally testing whether the incubation of recipient MDA-MB-231 cells with EVs stimulate clonal growth, the authors evaluated the migration of recipient IMR90 fibroblasts (Supp. Fig 9). Although these data are interesting, it is not clear why this EV-function readout (i.e. cell migration) is used at this stage. Does the incubation of MDA-MB-231 cells with EVs stimulate their clonal growth? Is ITGB3 in recipient cells required for this?

3. The link between syndecans and the mechanism of EV uptake that the authors described is still very weak. Perhaps the MS data for the proteome of EVs might inform on whether a particular syndecan is highly abundant in MDA-MB-231-EV preps, and if so, it would be of interest to evaluate the effect that knocking-down such syndecan in donor cells has on EV uptake.

4. The TITLE is misleading, as the authors did not study the integrin recycling throughout the MS. Instead, the phenotype regulating EV uptake is coordinated with integrin internalisation, and the title should reflect this.

Other points:

1. The order of presentation of the data and the organisation of the paper renders it difficult to read and understand. For example, line 181 pertains to figure 3c-d then the next figures that are mentioned are in the order of Fig2b, sup fig 8a, fig2e, fig 2f then figure 4a. Also Fig. 3b is not mentioned in text. This is similar in the supplementary data which should be in the same order in

the text and linked to the same data figure.

2. The choice to use IMR90 cell exosomes in figure 1 is not explicitly explained. The text needs a sentence or two to explain the reason for using these cells in relation to cancer.
3. Supplementary figure 6 needs to be labelled as such and in a separate data file
4. Supplementary figure 5A needs labelling control and shITGB3
5. The authors should discuss work from Meisner-Kober lab showing that filopodia mediate EV uptake (<https://rupress.org/jcb/article-lookup/doi/10.1083/jcb.201506084>)

Reviewer #2 (Remarks to the Author):

In the revised manuscript, the authors invested a large effort to further characterize the EV behaviour/fate at 4 degrees C (new fig2a and 3). The data and the study has been greatly improved. They also satisfyingly answers comment raised in point 2 of the initial review. Concerning the initial point 1 (interpretation of lack of surface binding at 4 C). This reviewer agrees that itg3 participates in EV uptake and is convinced by the data (especially new fig2a and 3). The explanations provided in the response letter are also clear. There is no strong EV binding at the cell surface at 4C. Again, this suggests (to this reviewer) the lack of BONA-FIDE EV-receptor because protein-protein interaction are normally maintained at 4 C. This reviewer feels that it should be clearly stated in the discussion that ITG3 participates in EV uptake but that the lack of surface binding at 4C suggests that ITG3 is not a bona fide EV receptor. This could be emphasised through comparison with the LDL receptor, using the same language used by the authors in the response letter.

To summarise, data are convincing, the model in fig 6 still stands especially the dynamin dependency, but the lack of strong statement discussing the point raised above is a high risk to mislead readers and the scientific community who may conclude that ITG3 is indeed a specific bona fide EV receptor.

Response to Reviewers' comments:

Reviewer #1 (Remarks to the Author):

This revision has significantly improved the paper and I appreciate the authors taking my suggestions on board. I feel, however, that some issues remain as the paper stands, particularly regarding the visualisation of internalised EVs in recipient cells by confocal microscopy.

We were pleased to see that the Reviewer was satisfied with the experiments carried out in the previous revision process and that he/she finds our manuscript significantly improved. We have now addressed the remaining issues guided by the Reviewers' suggestions.

Main points:

1. - Imaging

a) In order to overcome difficulties in distinguishing between donor and recipient ITGB3-positive structures using confocal microscopy (Supporting Figure 1), the authors could repeat these imaging experiments using EVs derived from ITGB3-knockdown cells. This could work as loss of ITGB3 in donor cells did not affect recipient cell-EV uptake (Figure 1). The experiment displayed in rebuttal letter Supporting Figure 3 could also be repeated using EVs derived from ITGB3-knockdown cells.

We are thankful for the Reviewer's suggestion to use EVs derived from ITGB3 knockdown cells to overcome the difficulties in distinguishing between donor and recipient ITGB3 positive structures. As suggested, we have repeated the imaging experiments in Supporting Figure 1 and observed a clear co-localization between ITGB3 and EVs at the cell surface (non-permeabilized conditions). These data are now included in new Supplementary Figure 6.

Furthermore, we performed co-localization experiments in permeabilized cells to visualize internal structures. As shown before in Supporting Figure 2, after 1h of treatment with PKH26-labeled EVs, the majority of vesicles is found in Lamp1 positive structures, highlighting the fast processing of EVs within recipient cells. These data are now included in Supplementary Figure 9. Under these experimental conditions, a significant co-localization between ITGB3 and shITGB3-derived EVs as well as ITGB3 and EEA1 was detected (Supplementary Figure 9). Thus, our co-localization studies further support the role of ITGB3 in the uptake of EVs.

b) In supporting Figure 4, the authors were not able to demonstrate the colocalisation between labelled EVs and p-FAK in recipient cells. Although we agree with the authors that confocal microscopy is at the limit of resolution for detection of single EVs, it is well established that EVs accumulate within the endosomal system upon internalisation in recipient cells, and EV-positive vesicular structures can be usually clearly visualised using such microscopy techniques. Are the PKH26-positive structures associated with focal adhesions? Live cell imaging of recipient cells transiently expressing GFP-FAK +/- ITGB3si in the presence of labelled EVs would help characterising the temporal events of FAK activation downstream of EV uptake, and to answer whether EVs co-localize with FAK in recipient cells.

We appreciate the Reviewer's proposal of alternative solutions for the co-localization studies with p-FAK and EVs. However, the suggested use of overexpressed GFP-FAK in live cell imaging experiments would only allow us to describe the intercellular localization of FAK but not to determine its activity state. We agree that in vivo imaging may allow to analyse a potential co-localization, however we thought that besides their co-localization, defining the activation of FAK by EVs is crucial to define the pathways involved. We provide evidence for the EV-dependent activation of endogenous FAK, using the pFAK-Y397 antibody as readout by Western blot analysis. Furthermore, we demonstrate the selective activation of FAK by the inclusion of the additional experiments requested in the first round of revisions (Supplementary Fig. 10). These data, combined with the functional studies on FAK activity for EV uptake (Figure 6) demonstrate that the selective activation of FAK by EVs is required for EV uptake.

c) In the rebuttal letter the authors include images of labelled EVs with ITGB3 and these images look a lot better than the ones shown in figure 3d. I think the other points made in the rebuttal letter should also be included in the paper. Also fig 3d would benefit from showing each channel separately black on white background as this would be much clearer.

We thank the Reviewer for his/her advice. We have now included the images and quantifications of EVs co-localized with ITGB3 and EEA1 (Supplementary Figure 9a) and EVs co-localized with EEA1 and Lamp1 in Supplementary Figure 9b, as described above. Furthermore, images are included (Figure 3c) with additional panels for the individual channels in black and white.

2. The authors show that depletion of EVs from CM impairs its ability to promote clonal growth in MDA-MB-231 cells (Fig. 1). However, instead of experimentally testing whether the incubation of recipient MDA-MB-231 cells with EVs stimulate clonal growth, the authors evaluated the migration of recipient IMR90 fibroblasts (Supp. Fig 9). Although these data are interesting, it is not clear why this EV-function readout (i.e. cell migration) is used at this stage. Does the incubation of MDA-MB-231 cells with EVs stimulate their clonal growth? Is ITGB3 in recipient cells required for this?

We appreciate the Reviewer's comment and have used the possibility to unify the readout for the function of EVs derived from WT and shITGB3 cells. We used EV-stimulated clonogenic growth as readout and determined the function of shITGB3-derived EVs in this process. Using the same experimental setup as in Figure 1, we determined the colony growth of WT and shITGB3 cells in the presence of conditioned media and EV-depleted conditioned media derived from shITGB3 cells. These experiments revealed that the stimulation of colony growth is independent of ITGB3 in EVs, but strictly reliant on ITGB3 in the EV-receiving cells. These data are now presented in the extended figure 1a and 1b. In line with our previous results, these data furthermore strengthen the concept that the function of cellular ITGB3, but not EV-localized ITGB3, is crucial for EV uptake and EV-stimulated colony growth. Finally, we would like to draw the Reviewer's attention to the fact that the data presented in supplementary Figure 9 are derived from indirect co-culture experiments as specified. We used double chamber plates with membrane inlet to allow the exchange of secreted factors like EVs from physically separated cells, and under these experimental conditions, the migratory capacity of IMR90 fibroblasts was stimulated when co-cultured with shCON but not with shITGB3 cells. These results are interesting, as the Reviewer pointed out, and might indicate differential functional effects of vesicle sub-populations depending on the phenotype studied – such as the stimulation of clonogenic growth by shITGB3-derived EVs but the inability of those EVs to stimulate cell migration in IMR90 cells. However, detailed future studies such as those carried out for the study of EVs in figure 1 are required to draw such conclusions. We would therefore prefer not to enter into detail in the main text, but, because of the interesting finding, we have kept the results as a supplementary figure.

3. The link between syndecans and the mechanism of EV uptake that the authors described is still very weak. Perhaps the MS data for the proteome of EVs might inform on whether a particular syndecan is highly abundant in MDA-MB-231-EV preps, and if so, it would be of interest to evaluate the effect that knocking-down such syndecan in donor cells has on EV uptake.

We agree with the Reviewer that the link between syndecans and the mechanism of EV uptake is not studied in detail. However, we would like to point out that our manuscript focuses on the role of ITGB3- and HSPG-modified proteins in EV uptake. For the role of HSPG-modified proteins, we provide compelling evidence with two different approaches in our manuscript (Fig. 2a,b and Fig 3b). As syndecans are the most prominent members of the family of HSPG-modified proteins and the interaction between ITGB3 and syndecans has been described, we speculated in the manuscript about the plausible link between syndecans and the mechanism of EV uptake. We have further modified the statement in the manuscript, describing "HSPG-modified proteins like syndecans" as an additional element required for ITGB3-dependent EV uptake. As suggested by the Reviewer, we have analyzed our MS data from the EV-proteome for the presence of highly abundant syndecans and other HSPGs. Out of the 17 described HSPGs, we have identified 11 differentially expressed HSPGs (including 3 syndecans) when comparing the EV proteome of WT and shITGB3 cells. As these data are derived from EVs, and it would be interesting to test in the future if this also holds true for the cellular abundance of HSPGs followed by knock-down experiments of the putative candidates alone and/or in combination. Taking into account the focus of our presented data, we believe that these studies are beyond the scope of the present manuscript.

Location	Description	Counts in experiments		Average Counts		log2FC
		shCTRL	shITGb3	shCTRL	shITGb3	
Membrane bound HSPGs	Syndecan-1 OS=Homo sapiens GN=SDC1 PE=1 SV=3 - [SDC1_HUMAN]	3	2	25.850	23.834	-2.016
	Syndecan-2 OS=Homo sapiens GN=SDC2 PE=1 SV=2 - [SDC2_HUMAN]	3	0	23.808	0	0
	Syndecan-4 OS=Homo sapiens GN=SDC4 PE=1 SV=2 - [SDC4_HUMAN]	3	3	25.054	23.540	-1.514
	Glypican-1 OS=Homo sapiens GN=GPC1 PE=1 SV=2 - [GPC1_HUMAN]	3	0	23.079	0	0
	Glypican-6 OS=Homo sapiens GN=GPC6 PE=1 SV=1 - [GPC6_HUMAN]	3	2	25.548	23.260	-2.288
	Neuropilin-1 OS=Homo sapiens GN=NRP1 PE=1 SV=3 - [NRP1_HUMAN]	3	3	26.587	25.299	-1.288
Secretory vesicles HSPGs	CD44 antigen OS=Homo sapiens GN=CD44 PE=1 SV=3 - [CD44_HUMAN]	3	3	28.564	27.913	-0.651
Extracellular matrix HSPGs	Serglycin OS=Homo sapiens GN=SRGN PE=1 SV=3 - [SRGN_HUMAN]	3	3	23.329	23.340	0.010
	Agrin OS=Homo sapiens GN=AGRN PE=1 SV=5 - [AGRIN_HUMAN]	3	3	22.574	22.375	-0.199
	Collagen alpha-1(XVIII) chain OS=Homo sapiens GN=COL18A1 PE=1 SV=5 - [COIA1_HUMAN]	3	0	23.519	0	0
	(Perlecan) Basement membrane-specific heparan sulfate proteoglycan core protein OS=Homo sapiens GN=HSPG2 PE=1 SV=4 - [PGBM_HUMAN]	3	0	20.986	0	0

4. The TITLE is misleading, as the authors did not study the integrin recycling throughout the MS. Instead, the phenotype regulating EV uptake is coordinated with integrin internalisation, and the title should reflect this.

We agree with the Reviewer's comment. Taking into account the title requirements for publication in Nature Communications, we propose to change the title to: ITGB3 mediated extracellular vesicle uptake facilitates intercellular communication to drive breast cancer metastasis.

Other points:

1. The order of presentation of the data and the organisation of the paper renders it difficult to read and understand. For example, line 181 pertains to figure 3c-d then the next figures that are mentioned are in the order of Fig2b, sup fig 8a, fig2e, fig 2f then figure 4a. Also Fig. 3b is not mentioned in text. This is similar in the supplementary data which should be in the same order in the text and linked to the same data figure.

We agree with the Reviewer. In the modified/updated version of the manuscript, we have reorganized figures 2 and 3. The data and figures are now presented in the correct throughout the text. The supplementary figures have been adjusted accordingly.

2. The choice to use IMR90 cell exosomes in figure 1 is not explicitly explained. The text needs a sentence or two to explain the reason for using these cells in relation to cancer.

The reviewer is right. We have introduced the use of IMR90 lung fibroblasts as a model for cells in the environment of the metastatic side in the lung. In the text we have included the following paragraph (line 126-133):

"During metastatic dissemination and homing within a different organ, neoplastic cells are exposed to EVs derived from a variety of cell types. In the metastatic mouse model, we have previously observed that the capacity of MDA.MB.231 cells to form metastatic colonies in the lung is reduced in shITGB3²⁶. We therefore asked if, besides MDA.MB.231-derived EVs, EVs from lung tissue cells were also able to stimulate colony growth of MDA.MB.231 cells. We therefore used lung-derived IMR90 fibroblasts: CM

from these cells, but not EV-depleted CM, was able to stimulate colony growth in MDA.MB.231 cells in an ITGB3-dependent manner (Figure 1a-b).”

3. Supplementary figure 6 needs to be labelled as such and in a separate data file

We apologize for the mistake and have corrected the labelling.

4. Supplementary figure 5A needs labelling control and shITGB3

We apologize for the omission of the labelling, which is now included in the corrected version of the figure.

5. The authors should discuss work from Meisner-Kober lab showing that filopodia mediate EV uptake (<https://rupress.org/jcb/article-lookup/doi/10.1083/jcb.201506084>)

We thank the Reviewer for pointing this out. We have now included the citation in the manuscript and discuss the described “endocytic hotspots” for EVs on filopodia in the context of integrin clustering and focal adhesion recycling (line 385).

Reviewer #2 (Remarks to the Author):

In the revised manuscript, the authors invested a large effort to further characterize the EV behaviour/fate at 4 degrees C (new fig2a and 3). The data and the study has been greatly improved. They also satisfyingly answers comment raised in point 2 of the initial review. Concerning the initial point 1 (interpretation of lack of surface binding at 4 C). This reviewer agrees that itg3 participates in EV uptake and is convinced by the data (especially new fig2a and 3). The explanations provided in the response letter are also clear. There is no strong EV binding at the cell surface at 4C. Again, this suggests (to this reviewer) the lack of BONA-FIDE EV-receptor because protein-protein interaction are normally maintained at 4 C. This reviewer feels that it should be clearly stated in the discussion that ITG3 participates in EV uptake but that the lack of surface binding at 4C suggests that ITG3 is not a bona fide EV receptor. This could be emphasised through comparison with the LDL receptor, using the same language used by the authors in the response letter.

To summarise, data are convincing, the model in fig 6 still stands especially the dynamin dependency, but the lack of strong statement discussing the point raised above is a high risk to mislead readers and the scientific community who may conclude that ITG3 is indeed a specific bona fide EV receptor.

We were pleased to see that the Reviewer was satisfied with the experiments carried out in the previous revision process and are very thankful for his/her constructive comments that have helped improve our manuscript.

Following the Reviewer’ suggestions, we have now addressed the remaining point and have included a clear statement in the manuscript that ITGB3 is not an EV receptor (line 360).

Reviewer #1 (Remarks to the Author):

Although I feel that this 2nd revision is an incremental improvement on the previous revision, there are still a number of questions that remain unanswered – such as the mechanism by which HSPGs affect EV uptake, or the identity of cellular compartments in which EVs accumulate in recipient cells. Furthermore, I feel that it is hard to draw any meaningful conclusions from some of the confocal microscopy data presented in the paper. For these reasons, I am still cautious to recommend this manuscript for publication in Nature Comms. as it currently stands.

Specific points:

- The title needs to be modified, as the paper does not address the effect of ITGB3-mediated EV internalisation on breast cancer metastasis. Even though previous work from the authors suggested that ITGB3 is required for breast cancer metastasis in the mouse lung, the title should only refer to work in the present manuscript. Instead of “breast cancer metastasis”, consider using “breast cancer clonal growth” or a similar term.
- Throughout the text, there seems to be confusion regarding the term “recycling of integrins”. In the trafficking field, “receptor recycling” refers to the “return” of receptors from intracellular compartments (e.g. recycling endosomes) to the plasma membrane. In many parts of the text (e.g. abstract line 38, introduction line 95, results line 204 and 307, discussion line 379), the authors use the term when instead should be referring to endocytosis/internalisation of integrins. This should be changed accordingly.
- The authors need to plot quantifications of EVs with sizes ranging 150 – 200 nm. Although the authors claim in the text that ITGB3-kd leads to a reduction of EVs within this range in the condition media, this is not shown in Fig 4C.
- The quantification of colocalisation between labelled EVs and recipient cell EEA1 (Supp. Fig. 9) revealed that the loss of ITGB3 in recipient cells does not influence the accumulation of EVs in early endosomes. Is it fair to say these observations do not follow the authors conclusions nor the model presented in Fig. 6H, where ITGB3 is suggested to mediate dynamin-dependent internalisation of EVs? Would one not expect the levels of EVs in early endosomes to be affected by ITGB3-loss? The observation that ITGB3-kd does not affect accumulation of EV in early endosomes needs to be added to the results text, and we advise the authors to discuss this in the discussion.
- Invert lookup table in the confocal microscopy images, so signal is displayed in black and background in white.
- Add in supplemental the FACS plots (I think that these had been in a previous version of the paper) from which the bar charts for EV internalisation come from. Also display how the gating was setup.
- Add citation in line 107 and line 300.
- Labels in Fig. 3C are wrong?
- For the story to flow in a more logical manner, we recommend that the authors move the data regarding the analysis of ITGB3-dependent EVs (Figure 4 and 5) to the end, following the requirement of FAK for EV internalisation, and change the text accordingly.

REVIEWER COMMENTS

Reviewer #1 (Remarks to the Author):

Although I feel that this 2nd revision is an incremental improvement on the previous revision, there are still a number of questions that remain unanswered – such as the mechanism by which HSPGs affect EV uptake, or the identity of cellular compartments in which EVs accumulate in recipient cells. Furthermore, I feel that it is hard to draw any meaningful conclusions from some of the confocal microscopy data presented in the paper. For these reasons, I am still cautious to recommend this manuscript for publication in Nature Comms. as it currently stands.

We were pleased to see that the Reviewer finds our manuscript significantly improved.

Our manuscript describes the functional and mechanistic role of ITGB3 in the process of EV uptake into the receiving cells. Starting from our previous results, describing the role of ITGB3 in breast cancer metastasis, we describe now for the first time, how integrin mediated uptake of EVs stimulates colony growth, and describe the mechanistic requirements, in particular the role of DYNAMIN, HSPGs and FAK, in this process.

Regarding the mechanism by which HSPGs affect EV uptake, we would firstly like to mention several previous studies (references 14, 15, 16 and 34 of the current version of the manuscript) demonstrating the physical interaction of HSPGs (particularly Syndecans) with EVs. Our study confirms those previous results and demonstrates that ITGB3 and HSPGs act within the same pathway of EV uptake (Fig.2A-B, Fig.6H). Secondly, we believe that the identification of a specific HSPG, as suggested by the Reviewer in the previous round, might provide deeper insights into the described mechanism but would not substantially improve the overall message of our manuscript. Furthermore, and as we have mentioned before (Main Points 3, revision1), we detected differential expression of 11 out of the 17 described HSPGs in our mass spectrometry analysis of MDA.MB.231 derived EVs, including the three members of the syndecan family. Functional redundancy among those HSPGs or the combinatorial effect of different HSPGs might hamper the identification of a specific HSPG and a detailed and extensive future studies will be required to address this issue in the future.

We have addressed the remaining issues following the Reviewers' suggestions and will particularly comment on the results and interpretation of the of confocal microscopy studies.

Specific points:

- The title needs to be modified, as the paper does not address the effect of ITGB3-mediated EV internalisation on breast cancer metastasis. Even though previous work from the authors suggested that ITGB3 is required for breast cancer metastasis in the mouse lung, the title should only refer to work in the present manuscript. Instead of “breast cancer metastasis”, consider using “breast cancer clonal growth” or a similar term.

We thank the Reviewer for his/her advice and have changed the title to “ITGB3-mediated uptake of small extracellular vesicles facilitates intercellular communication in breast cancer cells”.

- Throughout the text, there seems to be confusion regarding the term “recycling of integrins”. In the trafficking field, “receptor recycling” refers to the “return” of receptors from intracellular compartments (e.g. recycling endosomes) to the plasma membrane. In many parts of the text (e.g. abstract line 38, introduction line 95, results line 204 and 307, discussion line 379), the authors use the term when instead should be referring to endocytosis/internalisation of integrins. This should be changed accordingly.

We are thankful for pointing this out and have changed the term integrin recycling to integrin endocytosis or integrin internalisation in the parts of the manuscript referring to our own results (line 38, line 95, line 203, line 375).

- The authors need to plot quantifications of EVs with sizes ranging 150 – 200 nm. Although the authors claim in the text that ITGB3-kd leads to a reduction of EVs within this range in the condition media, this is not shown in Fig 4C.

Following the advice of the Reviewer, we have now included a fourth panel in Figure 4C with the quantification of EVs with sizes ranging from 150 – 200 nm. As already previously mentioned in the text, a significant reduction of EVs within this size range can be observed.

- The quantification of colocalisation between labelled EVs and recipient cell EEA1 (Supp. Fig. 9) revealed that the loss of ITGB3 in recipient cells does not influence the accumulation of EVs in early endosomes. Is it fair to say these observations do not follow the authors conclusions nor the model presented in Fig. 6H, where ITGB3 is suggested to mediate dynamin-dependent internalisation of EVs? Would one not expect the levels of EVs in early endosomes to be affected by ITGB3-loss? The observation that ITGB3-kd does not affect accumulation of EV in early endosomes needs to be added to the results text, and we advise the authors to discuss this in the discussion.

We acknowledge the comment of the Reviewer and the opportunity to discuss the results of Supp. Fig 9 in more detail. As suggest by the Reviewer in the previous round of review, we have included this Figure within the article, to demonstrate the co-localisation of ITGB3 with EVs and EEA1. We also agree with the statement of the Reviewer, that there is no alteration in the co-localisation between EEA1 and EVs in shITGB3 cells. As we have critically noted during the first round of review, for the interpretation of those results, the experimental conditions, particularly the resolution limit of the confocal microscope at approx. 100nm, need to be taken into account. This is particularly important, when considering the heterogeneity of EVs and the proposed role of ITGB3 in the selective uptake of small EVs. As we demonstrate in Figure 4 and 5, only small vesicles accumulate in the supernatant of ITGB3 knockdown cells (0-125nm, Fig 4C), arguing that especially the uptake of those vesicles is impeded. The abundance of bigger vesicles in the supernatant of in shITGB3 cells (125-500nm, Fig. 4C) is not significantly altered. In line with these results, no alterations in the portion of EVs detectable by confocal microscopy (> 100 nm) are observed (Supplementary Figure 9). We are therefore convinced that our observations follow our conclusions and to clarify this in our model, we included different sized vesicles in the schematic drawing and highlight EV heterogeneity and the role of ITGB3 in the selective uptake of small vesicles.

We also felt important to include the recent review by Kalluri et al (Science 2020) and extended our discussion as requested by the Reviewer. We highlight the relevance of EV heterogeneity and the current limitations of standard microscopic technics such as confocal microscopy to visualize individual small EVs at reliable resolution and/or to discriminate among different EV subpopulations *in vivo* (line 392 to 401 of the current manuscript).

Finally, it is important to note that the field of EV research is acknowledging during the last years the importance of EV heterogeneity in cancer and that our work is probably one of the first ones defining mechanistically this concept.

- Invert lookup table in the confocal microscopy images, so signal is displayed in black and background in white.

We thank the reviewer for his/her advice and present now the inverted images within the figures.

- Add in supplemental the FACS plots (I think that these had been in a previous version of the paper) from which the bar charts for EV internalisation come from. Also display how the gating was setup.

We have re-introduced the FACS plots as supplementary figures within the current version of the manuscript and have included additional representative examples of the applied gating strategies (Supplementary figure 16 to 19).

- Add citation in line 107 and line 300.

We are thankful for the advice of the Reviewer and have included now the references 26 (line 107) and 34 (line 300) within the manuscript.

- Labels in Fig. 3C are wrong?

We apologize for the mistake. The reviewer is right and we have corrected the labelling in Figure 3C.

- For the story to flow in a more logical manner, we recommend that the authors move the data regarding the analysis of ITGB3-dependent EVs (Figure 4 and 5) to the end, following the requirement of FAK for EV internalisation, and change the text accordingly.

We highly appreciate the efforts of the Reviewer to find the best possible solution for representing the data within the manuscript. Unfortunately, part of the experiments presented in Figure 6 (particularly Figure 6F) builds up on the experimental setting and results presented in Figure 4 and 5. To avoid the isolation of Figure 6F from the other results obtained for the requirements on FAK for EV internalisation, we would prefer to maintain the current order of the figures.